# Transposable elements contribute to cell and species-specific chromatin looping and gene regulation in mammalian genomes

Adam G. Diehl [ID] [1], Ningxin Ouyang[1] & Alan P. Boyle [ID] [1,2✉]

Chromatin looping is important for gene regulation, and studies of 3D chromatin structure across species and cell types have improved our understanding of the principles governing chromatin looping. However, 3D genome evolution and its relationship with natural selection remains largely unexplored. In mammals, the CTCF protein defines the boundaries of most chromatin loops, and variations in CTCF occupancy are associated with looping divergence. While many CTCF binding sites fall within transposable elements (TEs), their contribution to 3D chromatin structural evolution is unknown. Here we report the relative contributions of TE-driven CTCF binding site expansions to conserved and divergent chromatin looping in human and mouse. We demonstrate that TE-derived CTCF binding divergence may explain a large fraction of variable loops. These variable loops contribute significantly to corresponding gene expression variability across cells and species, possibly by refining sub-TAD-scale loop contacts responsible for cell-type-specific enhancer-promoter interactions.

---

[1] Department of Computational Medicine and Bioinformatics, University of Michigan, Ann Arbor, MI, USA. [2] Department of Human Genetics, University of Michigan, Ann Arbor, MI, USA. ✉email: apboyle@umich.edu

Ever since chromosomes were first observed microscopically, it has been speculated that their 3D structure plays a central role in regulating nuclear function[1,2]. Early observations revealed that individual chromosomes occupy distinct nuclear territories and, while their arrangement varies between different cell types, this structure is conserved between mother and daughter cells[2]. These findings led to the hypothesis that chromosome structure directly influences cellular phenotypes. Since that time, microscopic and molecular studies have dissected chromatin structure into an intricate hierarchy of large-scale territories, compartments, domains, neighborhoods, and loops[3–8], confirming the importance of 3D structure in regulating gene expression, replication, and other nuclear processes. However, the mechanisms by which these structures are created and maintained, and how they evolve are still poorly understood.

A common feature of chromatin loops is the presence of insulator proteins at their boundaries, most notably CTCF[3,4,9–14]. Although this property has been observed across distantly related metazoan phyla[10], it is especially important in mammals, where CTCF knockdown leads to widespread loop disruption and gene dysregulation[15]. This demonstrates that chromatin looping can be directly altered by changes in CTCF binding, with concomitant changes in gene expression[16–21], perhaps resulting from alternate enhancer–promoter contacts[22]. Comparisons between various human and mouse cell types have shown that differential looping is common[3,4,6], even between individual cells within a tissue[22], and there is evidence that CTCF-binding site divergence underlies this variation[23]. Differentially looped regions are associated with gene expression variation across both species and cell types in human, mouse, and chimpanzee[4,23,24], and are associated with aberrant gene expression in congenital diseases[25] and cancer[26]. Thus, CTCF divergence may explain phenotypic variations that are relevant to selection, serving as a possible catalyst for adaptive evolution.

TEs are mobile genetic elements that can influence observable phenotypes[27], often by altering the expression of nearby genes[28]. Many TEs proliferate through replicative mechanisms that enable exponential amplification and dispersal to new locations throughout their host genome. Importantly, TE's often carry embedded transcription factor (TF) binding sites (TFBSs) (reviewed in ref. [29]) and, in some cases, can be repurposed as tissue-specific enhancers[30,31]. Among TFs known to associate with TEs, CTCF is probably the best characterized, with many CTCF-binding sites deriving from TEs across multiple species[32–35]. While some TE-derived CTCF-binding sites may cause deleterious effects and be eliminated from the population, others may be maintained in the population as selectively neutral variants that may later be exapted for regulatory purposes, including chromatin loop anchoring. Many exapted loop anchors will be differentially present in the population, and those that influence gene expression may generate phenotypic variation upon which natural selection can act. Indeed, recent evidence has implicated TE proliferation in creating variation in CTCF binding associated with differential cancer risk among humans[36–39]. Furthermore, CTCF sites are enriched at the boundaries of chromatin loops in mouse immune cells[3] and human pluripotent cells[40]. Thus, it seems likely that looping variation may be an important mechanism by which TE activity elicits phenotypic changes. However, while a recent study demonstrated HERV-H elements at variable loop boundaries in human cells[40], the overall contribution of TE-derived CTCF sites to chromatin looping variation across mammals remains unclear. Furthermore, while some TE-derived CTCF sites appear to function in stabilizing conserved higher-order chromatin structures[41], the relative contribution of TEs to conserved and variable looping is also unknown.

In this study, we explore the contribution of TEs to CTCF binding and chromatin looping in the human and mouse genomes. We revisit the question of CTCF-binding enrichment in specific TE families and show that CTCF-binding enrichment appears to be a function of the strength of the CTCF motif within the TE consensus sequence. By contrast, functional exaptation appears to be a random process, with the likelihood of gain-of-function for any TE type depending mainly on its abundance. We demonstrate that the fraction of CTCF-tethered loops derived from TEs follows the fraction of TE-derived CTCF sites, showing that loop formation is not influenced by the evolutionary origin of the anchoring CTCF-binding site. We present evidence that TE-derived and non-TE-derived loop anchors (native loop anchors) are functionally and selectively equivalent, showing comparable levels of evolutionary constraint and similar patterns of activating histone marks. We further show that these functional signatures are significantly reduced in the non-looping sequence orthologs of species-specific loops.

We devised a system to classify loops by their breadth of use across cells and species and demonstrate that an association with looping variability is a general property of TE-derived CTCF-binding sites across cells and species. TEs are an important source of looping variability in human and mouse, with >50% of species-specific loops in mouse and >30% of cell-specific loops in human being TE-derived. Furthermore, this looping variation is significantly associated with gene expression variability across cell types and species. A subset of these variable loops appear to produce differential expression by refining regulatory neighborhoods to facilitate cell-specific enhancer–promoter interactions. We speculate that TE-driven CTCF-binding site expansions have contributed to looping diversity throughout human and mouse evolution, increasing the number of alternate chromatin conformations and, by extension, regulatory states a cell is able to adopt. This, in turn, may increase the flexibility of gene expression programs, thus enhancing adaptability in response to changing selective pressures.

## Results
Our ultimate goal was to investigate the impact of transposable element activity on chromatin looping divergence. To do so, we used publicly available ChIP-seq, ChIA-PET, and Hi-C data to identify chromatin loops in the human and mouse genomes, in which at least one anchor was derived from a transposable element. By classifying loops by their degree of sharing between species and cell types, we determined the contribution of TEs to cross-cell and cross-species looping divergence. We further investigated the properties and potential effects of TE-driven loop divergence by identifying sets of loops, in which a species-specific TE insertion created a differential loop (Fig. 1). We chose to focus on loops anchored by CTCF sites because of its well-characterized role in chromatin looping[3,5,11,14,15,19,42–49] and known enrichments within several families of SINE, LINE, and LTR retrotransposons in multiple mammalian species[32–34].

**TE-derived CTCF sites are prevalent in human and mouse.** We first assessed the genome-wide effects of TE proliferation on CTCF binding in human and mouse. CTCF ChIP-seq data for matched immune cell types from both species (Source Data File) were combined into a union set of orthologous and species-specific binding sites and intersected with known TE insertions[50] (Fig. 2a). The results show that TEs have contributed strongly to CTCF binding in both species, constituting ~35% of all CTCF-binding sites. CTCF binding was highly variable across species, with >85% of sites showing species-specific occupancy. TE-derived sites are overrepresented among species-specific CTCF-

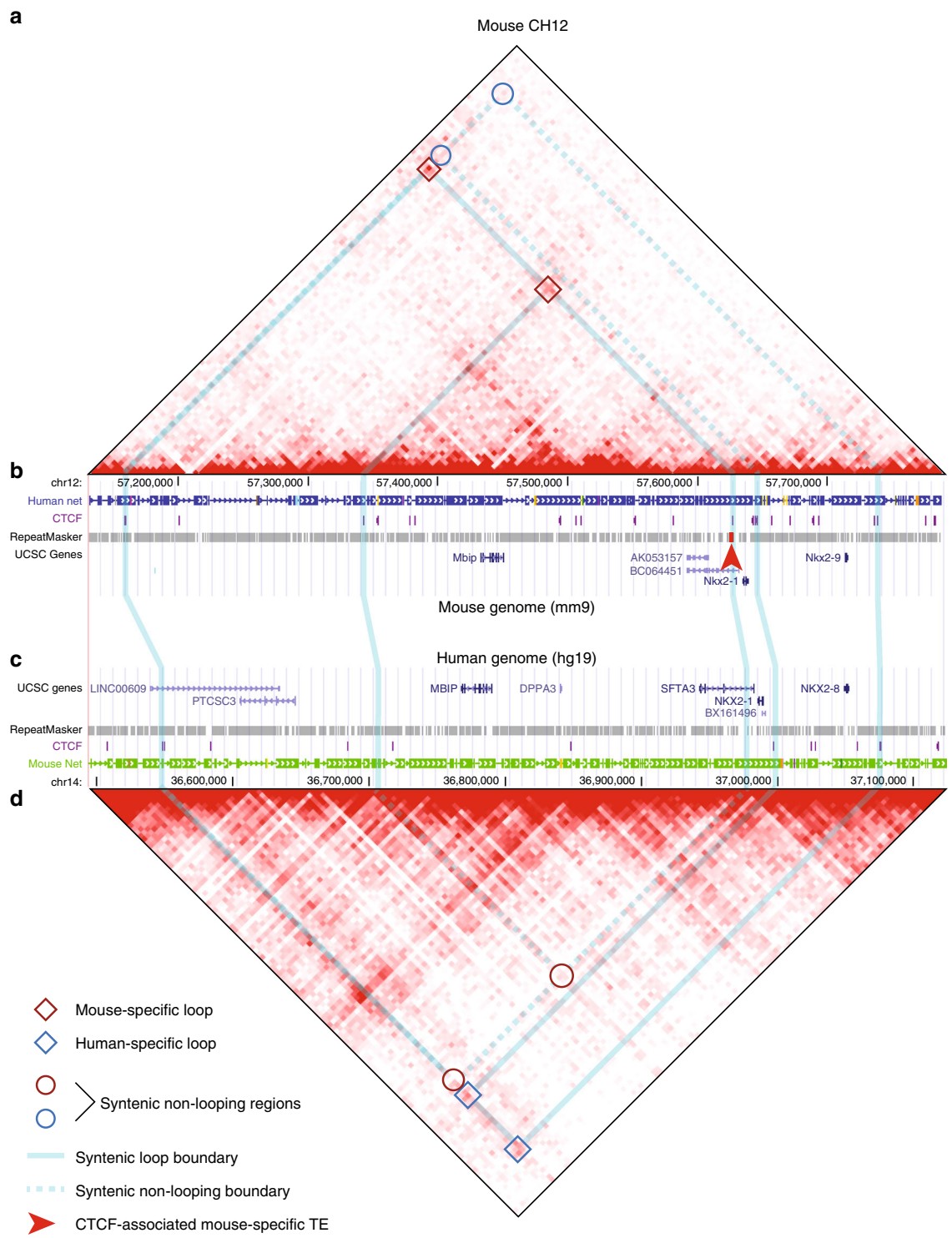

**Fig. 1 Transposable element insertions create novel species-specific loop contacts.** A differentially looped syntenic region of mouse chromosome 12 and human chromosome 14, in which the variable loop is anchored at a TE-derived CTCF-binding site. **a** Hi-C map of the region in mouse CH12 cells. Two mouse-specific loop contacts are indicated by dark red boxes, with their syntenic locations in the human genome indicated by red circles in (**d**). Blue circles indicate the location of human-specific loops in (**d**). **b** Relevant features of the mouse region in the UCSC Genome Browser. The right anchors of both mouse loops are tethered by a CTCF-binding site falling within a mouse-specific ERVK retrotransposon (bright red bar marked by arrowhead). **c** Relevant features of the orthologous region of the human genome in the UCSC Genome Browser. Syntenic locations of loop anchors observed in mouse and human are connected by vertical blue lines. **d** Hi-C map for the orthologous human region in GM12878 cells. Blue boxes indicate two human-specific loops, with their syntenic locations indicated by blue circles in (**a**). Red circles indicate the syntenic locations of mouse-specific loops in (**a**).

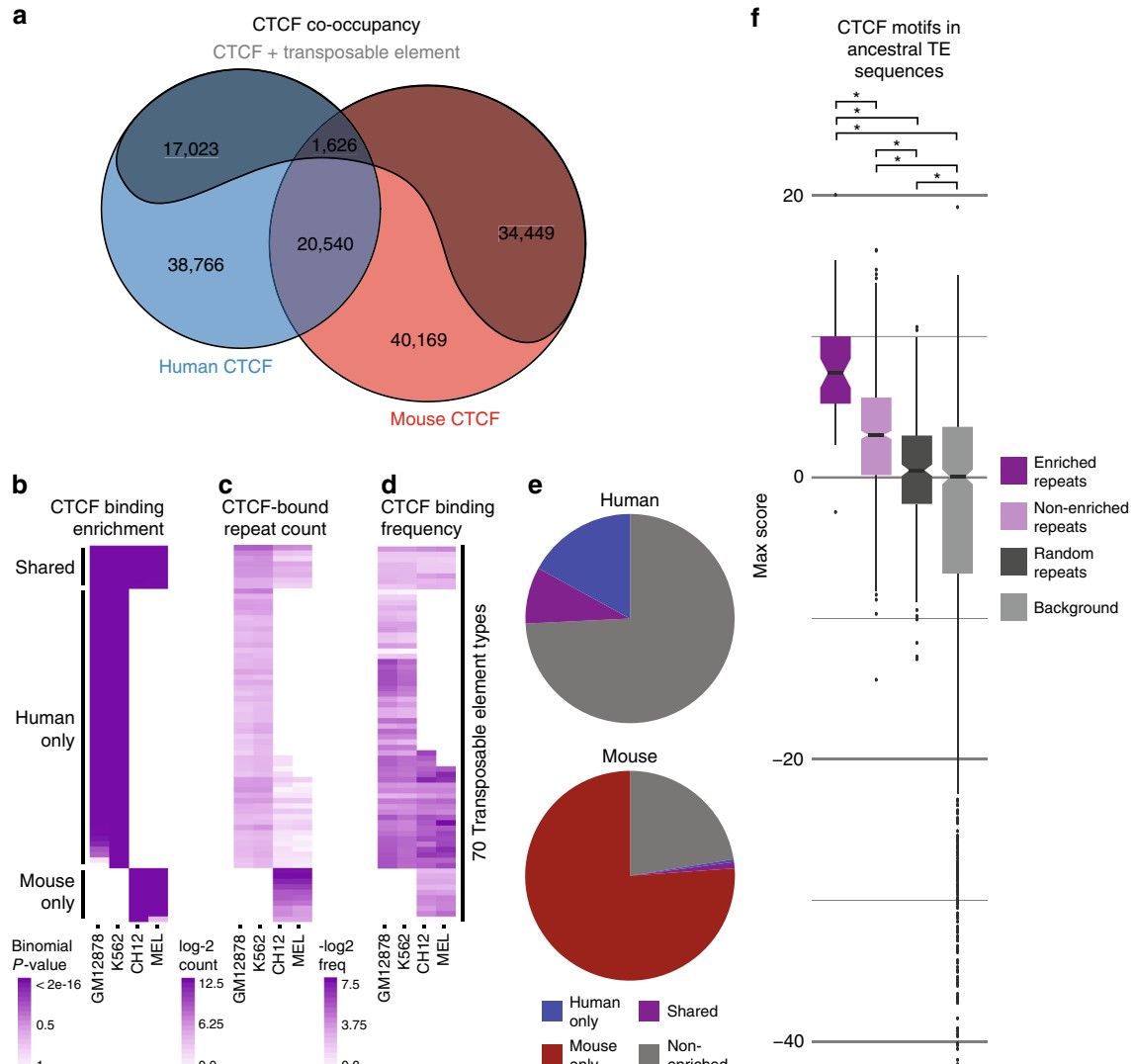

**Fig. 2 CTCF-binding variability is associated with transposable element activity. a** Proportion of CTCF sites in the human and mouse genomes with conserved and divergent binding and their respective transposable element (TE)-derived fractions. Human-specific and mouse-specific fractions include both orthologous and non-orthologous CTCF-binding sites. **b** Binomial tests recovered 70 TE types significantly enriched for CTCF binding. Enrichments were classified as human-only, mouse-only, or shared based on the cell types in which they were observed. **c** Cell-wise counts of CTCF-bound copies for each enriched TE type. **d** Cell-wise percentage of TE copies bound by CTCF for each enriched TE type. **e** Human and mouse fractions of TE-derived CTCF-binding sites originating from human-enriched, mouse-enriched, shared, and non-enriched TE types. **f** Log-odds score distributions for the strongest CTCF motif match within consensus of CTCF-enriched and non-enriched TEs, compared with TEs selected randomly from RepBase and length-matched background sequences. Scores above 1 represent sequences with greater than random resemblance to the CTCF motif. Enriched repeats, $n = 53$; non-enriched repeats, $n = 905$; random repeats, $n = 343$; background, $n = 958$. Boxplots are centered around the median, with upper and lower hinges indicating the first and third quartiles. Upper and lower whiskers extend from the hinge to the largest and smallest values within 1.5× the inter-quartile range from the hinge. Individual data points beyond the ends of the whiskers represent outliers. *One-sided Wilcoxon rank-sum test p-value < = 0.03.

binding sites, regardless of whether the insertion occurred before or after speciation, suggesting a link between TE activity and CTCF-binding divergence. Overall, TE-derived CTCF sites comprise >47% of mouse-specific sites (>36 times more than expected by chance) and >30% of human-specific CTCF-binding sites (>82 times more than expected by chance). These estimates are likely to be a lower bound given that many CTCF-binding sites may have originated from ancient repeats that can no longer be detected using current methods and many others may be filtered from the ChIP-seq data sets as non-uniquely mapped reads.

Previous studies have demonstrated strong CTCF-binding enrichments across multiple families of TEs in several mammalian species[32–34]. In our analysis, we identified many previously unreported TE types contributing substantially to CTCF binding

in human and mouse. This prompted us to revisit CTCF-binding enrichments within our data set, given that it includes a larger sampling of sites across previously untested cell lines. We adapted enrichment testing methods used in three previous studies[32–34]. Overall, we observed good agreement between the results from all three methods (Supplementary Fig. 1, Supplementary Tables 1–3) and our results, which capture over 76% of previously reported CTCF enrichments. Therefore, CTCF-binding enrichments are highly robust and reproducible, despite our use of different cell types. Because the binomial testing method produced the most conservative results, we chose to use these data for subsequent analyses.

Binomial testing yielded 70 CTCF-enriched TE types (Fig. 2b; Supplementary Table 1). We detected enrichments that were

specific to mouse (mouse-only types), specific to human (human-only types), and a small set of TE types enriched in both species (shared types). It is important to note that human-only and mouse-only types are categorized based purely on CTCF-binding enrichments, thus human-only and mouse-only TE types may be either species-specific or ancestral in their amplification patterns. Enrichment strength was correlated with neither TE abundance (Fig. 2c) nor CTCF-binding frequency (Fig. 2d), and all but a few enrichments spanned both cell types within a species, showing that CTCF-binding enrichments are not cell-type specific (Supplementary Fig. 1C–D). Our methods recovered 60 previously unobserved TE types, likely due to the larger size of the present data set compared with previous studies. All TE types in which we observed enrichments were previously identified in a recent genome-wide screen for regulatory exaptation of TE elements in human[51]. However, whereas that study identified exaptation events from all known families of TEs, we saw only a subset of these in our data: L1 LINEs; Deu and B2 SINEs; ERV and Gypsy LTRs; hAT and tcMar DNA elements, and two types of mouse-specific L1-dependent retrotransposons (Supplementary Table 1).

While these observations agree well with previous reports, we also observed that a large fraction of both human and mouse CTCF-binding sites fall within instances of TEs from families not enriched for CTCF binding (Fig. 2e). In fact, all but two of the major classes (LINE_Merlin and DNA_Dong-R4) found by Haussler and Lowe[51] were represented in our data set, even though only a subset of these were CTCF-enriched. Given the robustness of our tests, we are confident that non-enriched types do not represent false negatives, leading us to speculate that functional exaptation does not reflect any properties unique to CTCF-enriched TE types, but rather the presence of a suitably strong CTCF-binding motif. Consistent with this hypothesis, we observed that CTCF motifs are present in the consensus sequences (a proxy for the ancestral TE sequence) for all enriched and non-enriched TE types (Fig. 2f), as well as many others: in total, 38% of human and mouse consensus sequences in Repbase[52] contain CTCF motifs.

Notably, observed trends in CTCF-binding strength, motif scores in individual motif instances, and phastCons conservation scores between enriched and non-enriched TE types do not appear to explain differences in enrichment (Supplementary Fig. 2). This suggests that enrichments may primarily reflect the presence of a strong CTCF site within a TE type's consensus sequence. This may allow newly inserted copies of enriched TEs to divert CTCF from nearby non-TE binding sites (native sites) more efficiently than novel copies of non-enriched TEs. By contrast, the chance of exaptation likely depends on extrinsic factors such as location, local chromatin environment, and proximity to functional sequences including genes, cis-regulatory elements, and existing CTCF-binding sites. Therefore, enrichment does not appear to be a prerequisite for exaptation.

**CTCF enrichments reflect differential TE exaptation.** Previous reports noted coincidence between TE dispersals and primate divergence dates[30,53]. We wanted to see if CTCF-enriched TEs might show a similar association with human–rodent divergence. TE insertion ages were estimated for all enriched TE types using previously published methods[32], and score distributions were plotted relative to the estimated primate–rodent divergence date, ~75 million years ago (Fig. 3a). We noted that the majority of human-specific and mouse-specific TE types had median dispersal dates within 25 million years of the estimated primate–rodent divergence date. The only exceptions in mouse were the B2 sines and IAPEY4_LTR, a type of mouse-specific

ERV2 LTR. In human, the exceptions all belonged to human-specific members of the ERV1 and ERV2 classes.

Intriguingly, we noticed that the age distributions for most human-only TE types indicated amplification predated primate–rodent divergence by a significant margin. In fact, nearly half of human-only CTCF-binding enrichments were observed in ancestral TE types, which were amplified primarily in the most recent common ancestor (MRCA) of rodents and primates[54]. For discussion purposes, we call these HOA types, for (H)uman-(O)nly (A)ncestral. All but five HOA types showed evidence for CTCF binding in mouse, and had human and mouse age distributions consistent with a single dispersal in the MRCA. This led us to wonder why HOA types only showed binding enrichments in human. We hypothesized that, although HOA TEs were amplified in the MRCA, differential exaptation in mouse and human led to the observed species-specific enrichments. In this scenario, the CTCF motifs within mouse orthologs of TE copies exapted in human would be under relatively less functional constraint, and thus would decay over evolutionary time and eventually lose the ability to bind CTCF. However, because we noted expanded human copy number for many HOA types (Fig. 2c), and six HOA types showed age distributions consistent with human-specific amplification after divergence, we wanted to rule out the possibility that HOA enrichments result from primate-specific TE dispersals after divergence from rodents.

To investigate this possibility, we used mapGL[55] to label TE-derived CTCF sites as orthologous (i.e., present in both genomes), lost from the mouse genome, or gained in the human genome (Fig. 3b). If enrichments primarily reflect human-specific TE amplifications, we would expect most copies of HOA TEs to be labeled as human gains. As an internal control, we note that human-specific TE types and mouse-specific TE types all show the expected distributions dominated by species-specific gains. However, orthologs and sequences lost from the mouse genome after speciation constitute the overwhelming majority of HOA loci, with proportional contributions closely matching those seen for shared TE types. The only exceptions were the six previously mentioned HOA types for which age distributions suggested human-specific amplifications. For these six types, we cannot rule out that human-specific amplification has influenced the observed CTCF-binding enrichments. However, it is reasonable to conclude that this mechanism cannot explain enrichments for the majority of HOA types.

To verify our hypothesis that only human copies of HOA types have retained the ability to bind CTCF, we compared the maximum human and mouse motif scores for all HOA instances that were both occupied by CTCF in human and mappable to the mouse genome (Fig. 3c). As expected, mouse instances scored systematically lower than their human counterparts and >2/3 had maximum motif scores below one, the theoretical score threshold for CTCF binding. By contrast, the majority of human instances had maximal scores consistent with robust CTCF-binding ability. While the consensus sequences for all these TEs contain strong CTCF motifs, these data show that mouse instances have progressively lost the ability to bind CTCF, explaining the observed differential enrichments.

**TE-derived and native CTCF sites are functionally equivalent.** We next wanted to determine the contribution of TE-derived CTCF sites to human and mouse loop anchors. RAD21 ChIA-PET loops from human GM12878 and K562 cells and Hi-C loops from mouse CH12 cells (Source Data File) were filtered for CTCF presence and trimmed to coincide with the strongest embedded CTCF ChIP-seq peak at each anchor, then intersected with TE

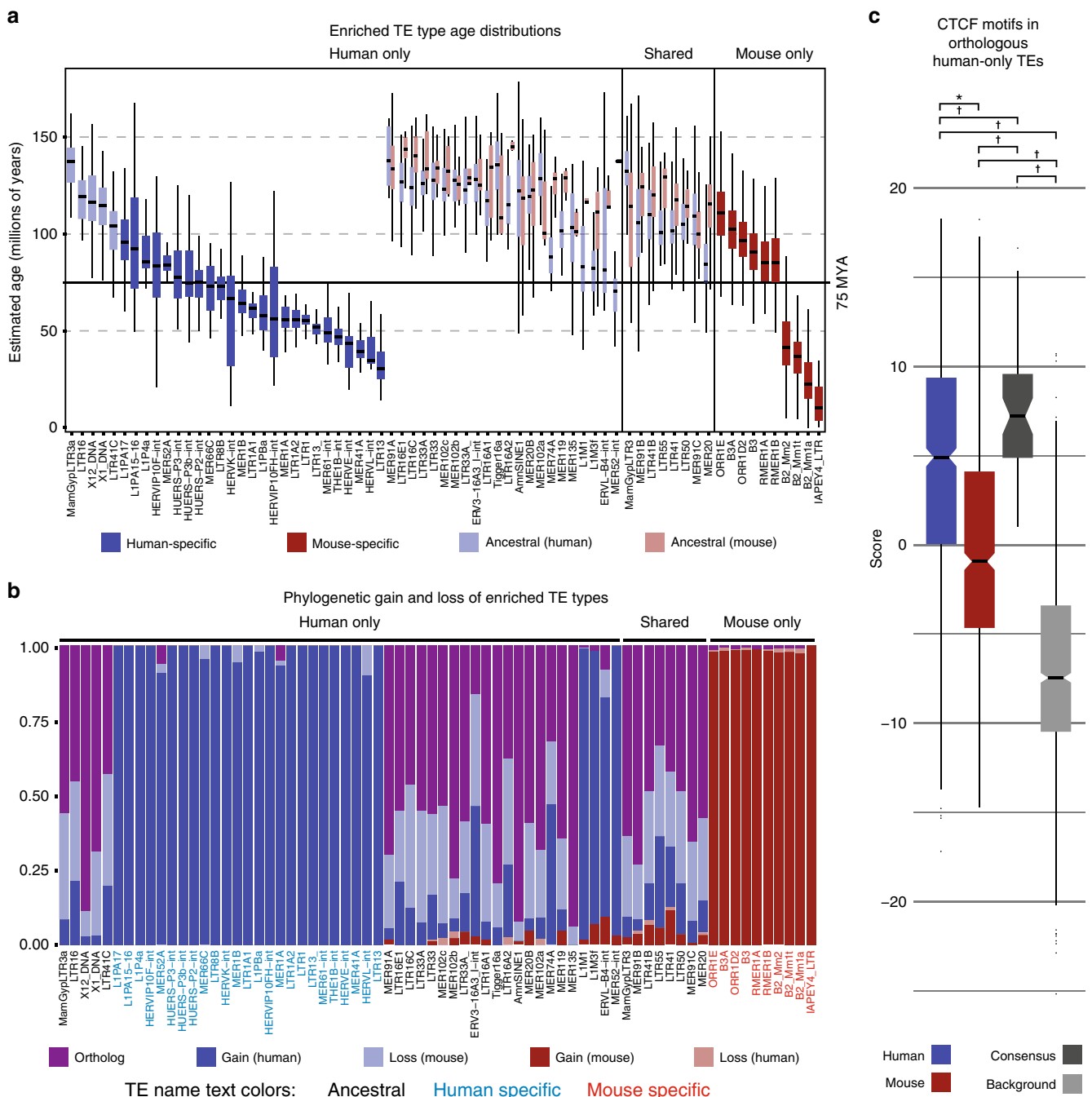

**Fig. 3 Ages and phylogenetic histories of human-only CTCF-enriched TEs support mouse-specific loss-of-function. a** Estimated age distributions for CTCF-bound TE insertions of enriched types. Colors indicate the species-specificity of each TE type and score distributions are split by species where applicable. The solid black horizontal line marks the estimated primate–rodent divergence date. **b** Fraction of enriched TE insertions inferred as orthologous, or as evolutionary gains or losses on a given branch of the phylogeny. TE label colors indicate whether the given type is ancestral (black), human-specific (blue), or mouse-specific (red). **c** Maximum CTCF motif scores within human and mouse instances of ancestral, human-only TE types. TE Consensus sequences (a proxy for the ancestral TE sequence, dark gray) and length-matched background sequences (light gray) are shown for comparison. Log-odds scores greater than 1 indicate sequences matching the CTCF motif more than expected by chance. Hypothesis tests were used to assess significance of differences in all pairwise comparisons: *$p <= 2.9e-58$ (one-sided Wilcoxon signed-rank test), †$p <= 2.6e-4$ (one-sided Wilcoxon rank-sum test). Human, $n = 852$; mouse, $n = 852$; consensus, $n = 54$; background, $n = 852$. Boxplots in **a** and **c** are centered around the median, with upper and lower hinges, indicating the first and third quartiles. Upper and lower whiskers extend from the hinge to the largest and smallest values within 1.5× the inter-quartile range from the hinge. Individual data points beyond the ends of the whiskers represent outliers.

annotations[50]. We observed that TE-derived CTCF-binding sites were present at ~15% of loop anchors. Therefore, we would expect 27.9% of all loops to include at least one TE-derived anchor, with ~2.3% derived from two TE-derived anchors, if TE-derived and native loop anchors are functionally equivalent. In

fact, the fractions we observed matched these expectations almost exactly: 25.1% of loops included one TE-derived anchor (Fig. 4a) and 2.6% were formed from two TE-derived anchors (Fig. 4b). These fractions closely parallel previously reported TE-derived contributions to looping[41]. Furthermore, ~88–89% of both TE-

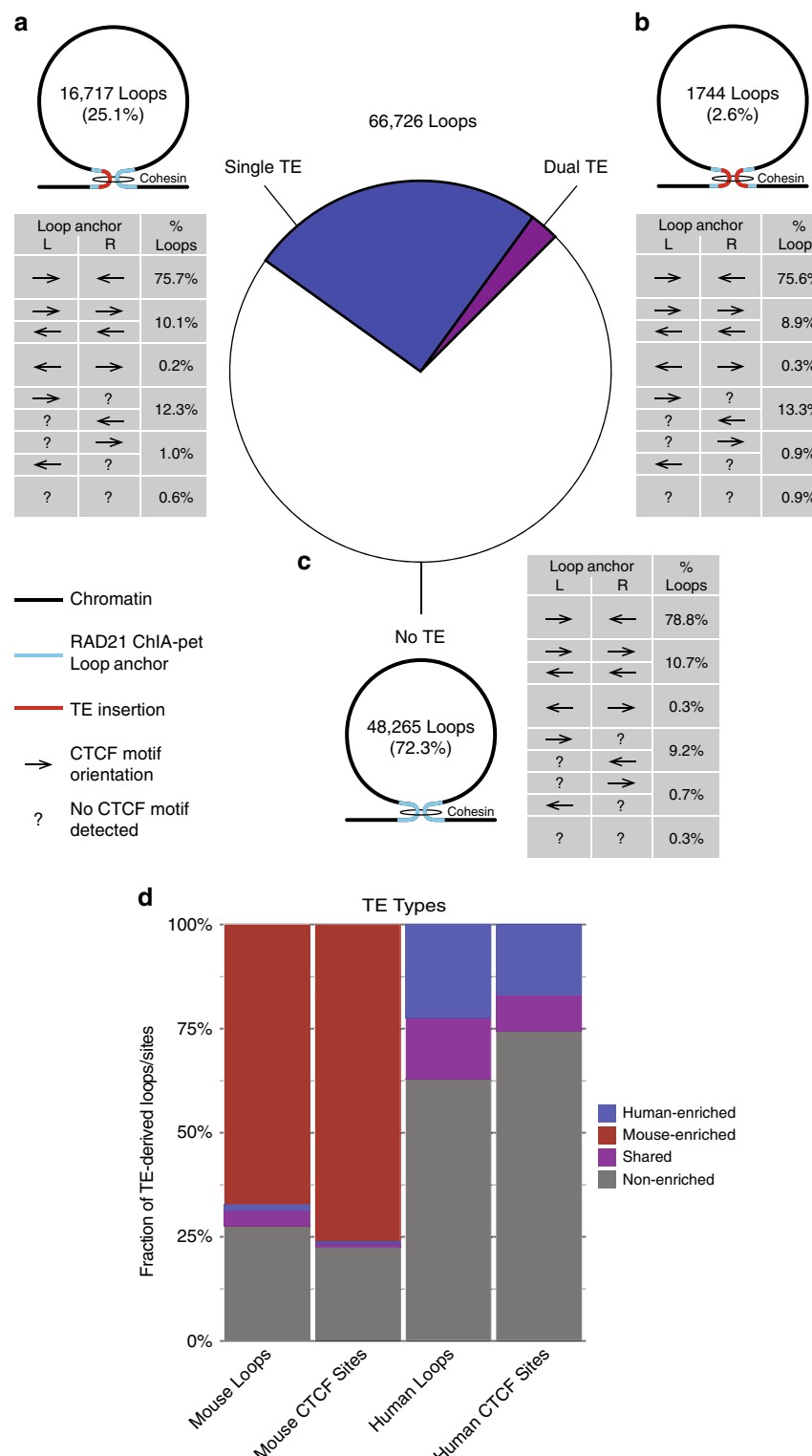

**Fig. 4 Transposable elements and native CTCF-binding sites form functionally equivalent chromatin loop anchors.** Human RAD21 ChIA-PET and mouse Hi-C loops containing CTCF ChIP-seq peaks at both anchors are broken down according to the number of TE-derived anchors they include. Each possible configuration is shown as a pictograph, with associated counts, alongside the central pie chart showing the fraction of all loops contributed by each configuration. Tables show the prevalence of different CTCF motif arrangements for each loop configuration. **a** Prevalence and CTCF motif arrangements for loops formed between pairs of TE-derived and native anchors. **b** Prevalence and CTCF motif arrangements for loops formed between pairs of TE-derived anchors. **c** Prevalence and CTCF motif arrangements for loops formed between anchors not derived from known TE insertions. **d** Contributions of CTCF-enriched and non-enriched TE types to TE-derived loops and CTCF-binding sites in human and mouse.

derived and native loops contained pairs of convergent or inward-pointing CTCF motifs, recapitulating known properties of chromatin loops[3,7,48], and these fractions were stable across all three cell types (Supplementary Fig. 3). Finally, both TE-derived and native loops exhibit similar CTCF-binding properties (Fig. 4a–c) and patterns of activating histone marks (Supplementary Fig. 4). Altogether, these observations strongly support functional equivalence of TE-derived and native chromatin loop anchors.

Consistent with our hypothesis that CTCF-enrichment does not directly influence exaptation, we found that the fractions of enriched and non-enriched TE types contributing to human and mouse chromatin loops are proportional to their overall contributions to CTCF binding (Fig. 4d). Indeed, all but nine of the TE families present in our CTCF data set have contributed chromatin loop anchors in human and/or mouse, and their overall contributions roughly scaled with their abundance. This strengthens our conclusion that exaptation requires only a suitably strong CTCF motif, and that, once bound by CTCF, cooption as a chromatin loop anchor results from the same mechanisms that determine pairing between native anchors.

**Sub-TAD and TAD scale chromatin loops are highly variable.** To further explore the evolutionary significance of TE-derived CTCF-binding sites on chromatin looping, we wanted to quantify the contributions of exapted CTCF-binding sites to conserved and variable loops across these cell types. We designed an algorithm to classify loops based on their degree of conservation across cells and species (Supplementary Fig. 5). Seven discrete conservation classes were defined based on cross-species mappability and overlap between annotated loop anchors in pairs of query and target cells (Fig. 5a), allowing us to describe chromatin looping conservation in much greater detail than previously possible and quantify the contribution of exapted loop anchors across a spectrum of conservation levels.

Our results showed a high degree of chromatin looping variability between all possible pairwise combinations of cell types, both within and across species (Supplementary Fig. 6). Less than 25% of loops were fully conserved between human and mouse, and only ~50% of loops were fully conserved between GM12878 and K562 cells (Fig. 5b). These rates are substantially lower than previous reports of 55–84% loop conservation across human cells, and 45–76% between human and mouse[3,4]. However, we used a more stringent definition of conservation stemming from two key differences between our methods and those used in previous studies. First, previous methodologies required only that both query loop anchors map to any pair of anchors in the target cell, regardless of whether that pair forms a coherent target loop. Approximating this definition by aggregating data from classes C, B2, and B1 yields conservation estimates very close to those reported previously[4]. Second, our resolution for detecting differential loop contacts was much higher than previous studies because ChIA-PET data were trimmed to coincide with the strongest embedded CTCF ChIP-seq peak, and these coordinates were not extended in cross-mapping analyses. This raises an important point in comparing chromatin looping across cells, tissues, and developmental stages: the scale at which loop divergence may yield biologically meaningful outcomes, and therefore the appropriate resolution for this type of analysis, is currently unknown. However, we note that, while extending the window around the CTCF-binding peak by 10–50 kb increased the overall degree of loop conservation, bringing it in line with levels previously published[4,41], it had negligible effects on TE contributions to conserved and non-conserved classes (Supplementary Fig. 7).

Previous studies have reported that strong conservation across cells and species is a hallmark of topologically associating domains (TADs)[3,4]. With this in mind, we separated known TADs from the rest of our data set and plotted their conservation classes (Fig. 5c). While we expected a strong enrichment of conserved classes, we found only modest overrepresentation of class C loops among TADs. While ~60% TAD conservation between human cell types is in line with Dixon et al.[4], we saw human–mouse conservation of only ~30%—roughly half the level observed by Dixon et al.[4]. The most-likely explanation seemed to be different definitions of conservation, most notably our requirement that both loop anchors in query loops coincide with a single target loop while their methods require only overlap between individual domain boundaries. Using their definition yielded comparable results for cross-species conservation (Supplementary Fig. 8). These results are consistent with the recently proposed model of TADs as dynamic structures with a high degree of variability between species, cell types, and individual cells[8,24].

**Looping divergence is correlated with functional divergence.** If conservation classes accurately capture the functional conservation of chromatin loops, we would expect to observe a correlation between conservation classes, evolutionary conservation, and other functional annotations. Indeed, conservation classes appear to reflect underlying phylogenetic conservation within loop anchors, visible as a well-defined peak of PhastCons[56] scores centered at the CTCF-binding site which declines in magnitude with decreasing loop conservation (Fig. 5d). This pattern remained evident even when only TE-derived loops were considered (Supplementary Fig. 9). This pattern may be partially explained by the younger evolutionary age of many species-specific loop anchors and decreased constraint on divergent ancestral elements. However, even the least conserved classes of loops show strong evidence for functional constraint. This may reflect the fact that the exapted loop anchors analyzed are restricted to those that bind CTCF and form chromatin loops sufficiently strongly as to be detected by ChIA-PET. Therefore, it is possible that variable loop anchors contributing to loops below the detection threshold of this analysis are subject to lower levels of functional constraint. We also noted correspondence between cell-specificity of activating histone mark patterns and conservation classes, with stronger enrichments for activating histone marks evident in cells in which a loop has been observed at a given locus compared to those without an annotated loop (Supplementary Fig. 10). Thus, we conclude that structural divergence introduced by TE-derived loops is indeed associated with functional divergence.

**Exapted CTCF sites contribute strongly to variable looping.** We next examined the relative contributions of exapted loop anchors to each conservation class. A recent report showed that TE-derived CTCF site have stabilizing effects on chromatin looping[41]. If this is the primary function of these sites, we reasoned that TE-derived anchors should be most prevalent among conserved loops. However, since TEs have previously been associated with variable looping and gene expression[40], an association with non-conserved loops may also be expected. Our results show that TE-derived loops play roles in both mechanisms, contributing substantially to loops in all conservation classes. However, their abundance increases as conservation decreases (Fig. 6a–c), showing that they contribute relatively more to looping variability than conservation. Surprisingly, this was true in both cross-species comparisons (Fig. 6a, b) and in intraspecies comparisons between human cell types (Fig. 6c).

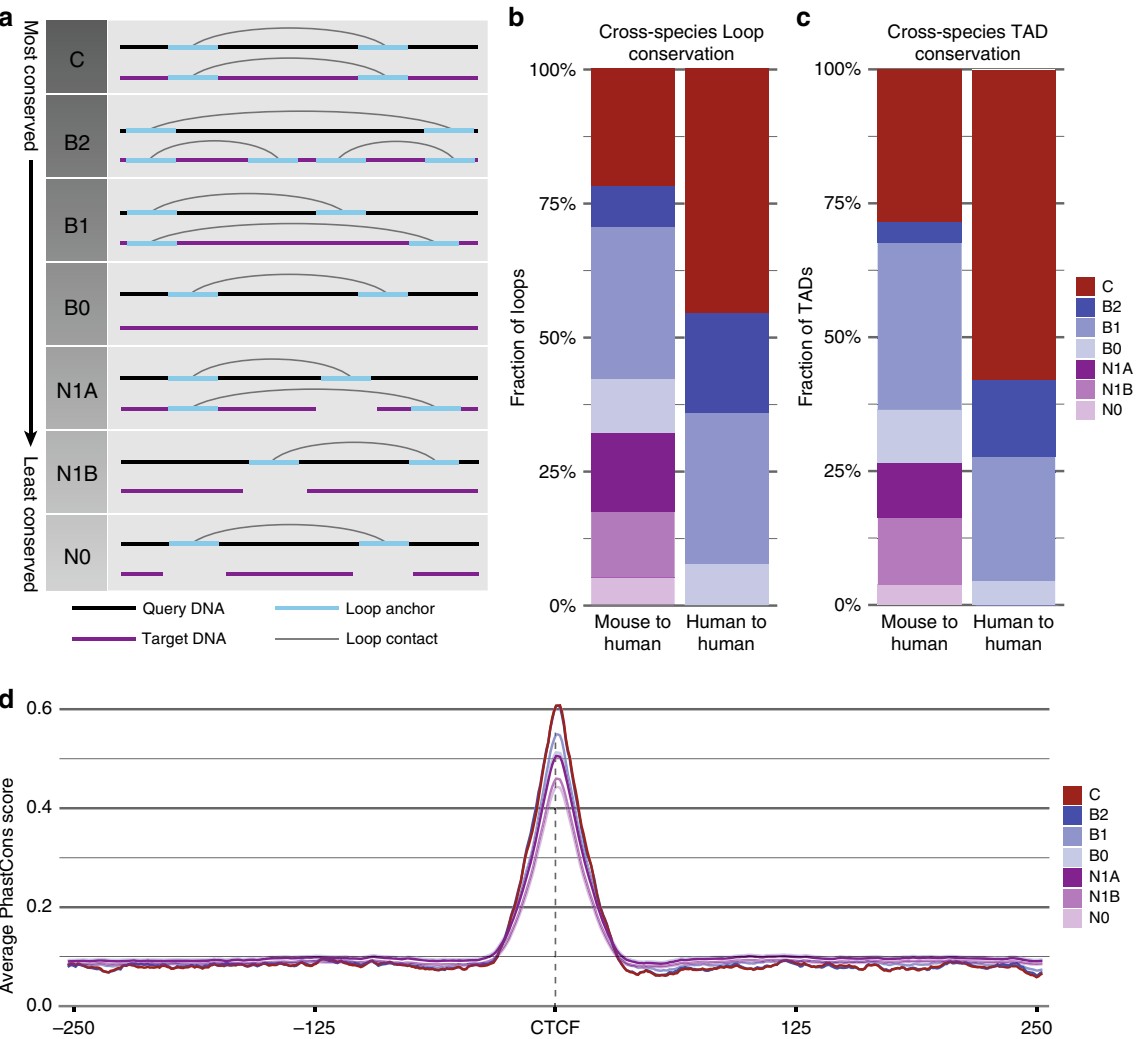

**Fig. 5 Conservation classes describe loop co-occurrence patterns across cells and species, and are correlated with underlying phylogenetic conservation. a** Conservation classes describe varying degrees of loop conservation between a query cell and a target cell based on sequence mappability and presence of a loop anchor at the syntenic locus. Diagrams illustrate the possible arrangements when comparing a query loop to a syntenic region in the target cell, with their corresponding conservation class labels. **b, c** Conservation class assignments from all pairwise comparisons between different cell types were aggregated into human–human and mouse–human comparisons. **b** Contributions of each conservation class to all chromatin loops. **c** Contributions of each conservation class to the subset of loops corresponding to known TADs. **d** Average phastCons conservation scores in 500 bp windows centered at the CTCF ChIP-seq peak summit within human GM12878 loop anchors plotted for each conservation class, with CH12 used as the target cell.

Indeed, the trend was present in all pairwise cell comparisons (Supplementary Fig. 11), and was not affected by decreasing the resolution of loop comparisons (Supplementary Fig. 12).

In mouse–human and human–human comparisons, we see a trend toward larger contributions of species-specific enriched TE types with decreasing loop conservation, particularly in the "N" conservation classes (Fig. 6d, e), and a trend toward younger TE ages in the same classes (Fig. 6g). This is not surprising and may be easily explained by a combination of continued TE activity after divergence and differences in exaptation frequencies we observed. However, the same TE types are uniform in their contributions and age distributions across conservation classes in human–human comparisons (Fig. 6f, g). This demonstrates that the association with variability cannot be explained by neutral selective forces, differential TE content, nor differential CTCF enrichments alone.

**TE-derived looping correlates with variable gene expression.** We wanted to investigate a possible correlation between the looping variability we detected and variable gene expression. Because it is notoriously difficult to reliably assign distal regulatory elements to their target genes, we first wanted to isolate a subset of loops representing likely enhancer–promoter interactions. We first verified the presence of enhancer–promoter contacts by annotating loop anchors in each cell type with H3K4me3 and H3K4me1 signal. This allowed us to identify individual loop anchors as likely enhancers or promoters and classify loops as enhancer–promoter, promoter–promoter, and enhancer–enhancer interactions (Supplementary Fig. 13A). We annotated loop anchors with their nearest gene based on distance to the transcription start site (TSS) and used TSS distance to isolate loops between one promoter-proximal anchor (< = 1 kbp) and one distal anchor (> = 3 kbp). The difference in gene expression (ΔTPM) for the gene nearest to the promoter-proximal loop anchor was calculated across pairs of query and target cells for TE-derived and native loops in conserved and non-conserved subsets of the data set. Any association between loop variation and variable gene expression should be apparent as

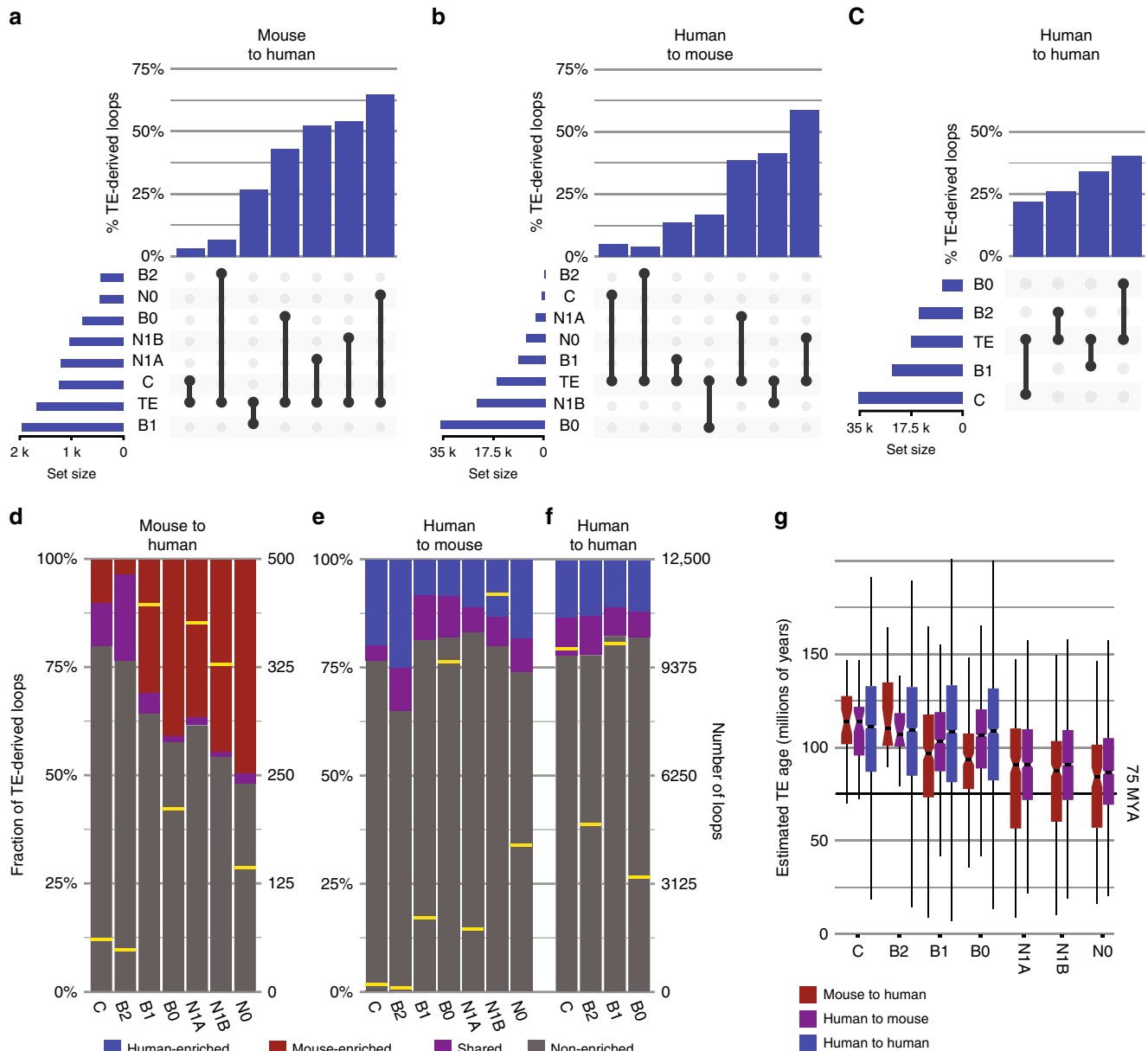

**Fig. 6 Transposon-derived loop anchors contribute disproportionately to species-specific and cell-specific loops. a** UpSet plot[77] showing TE-derived fractions within each conservation class for the comparison of mouse query loops to human target loops. Horizontal bars show the number of loops assigned to each conservation class and the observed number of TEs among query loops. Vertical bars show the fraction of loops derived from TEs within each conservation class, ordered from left to right by decreasing structural conservation. **b** Same as (**a**), but between human query loops and mouse target loops. **c** Same as (**a**) and (**b**), but for bidirectional pairwise comparisons between human GM12878 and K562 loops. **d** Fraction of TE-derived loops contributed by CTCF-enriched TE types in each conservation class for aggregate data from mouse–human comparisons. Yellow bars indicate the number of loops observed in each conservation class (right scale). **e** Same as (**d**), but for aggregated data from human–mouse comparisons. **f** Same as (**c**) and (**d**), but for aggregated data from human–human comparisons. **g** Age distributions of TE insertions found at loop anchors in each conservation class for mouse–human (red), human–mouse (purple), and human–human (blue) comparisons. The estimated rodent–primate divergence date (75MYA) is indicated by a bold horizontal line. Boxplots are centered around the median, with upper and lower hinges, indicating the first and third quartiles. Upper and lower whiskers extend from the hinge to the largest and smallest values within 1.5 × the inter-quartile range from the hinge. Individual data points beyond the ends of the whiskers represent outliers. C: mouse–human, $n = 37$; human–human, $n = 6374$; human–mouse, $n = 58$. B2: mouse–human, $n = 26$; human–human, $n = 3106$; human–mouse, $n = 20$. B1: mouse–human, $n = 386$; human–human, $n = 6126$; human–mouse, $n = 1149$. B0: mouse–human, $n = 223$; human–human, $n = 2029$; human–mouse, $n = 5200$. N1A: mouse–human, $n = 394$; human–mouse, $n = 996$. N1B: mouse–human, $n = 416$; human–mouse, $n = 7755$. N0: mouse–human, $n = 170$; human–mouse, $n = 2457$.

a measurable shift in ΔTPM values between conserved and variable loop classes. Indeed, we saw significant shifts toward greater ΔTPM in variable loops in all but one comparison in aggregated mouse–human and human–human data sets for both TE-derived and native loops (Fig. 7a) and across all individual pairs of cells (Supplementary Fig. 13B-C). The sole exception was for TE-derived loops in the mouse–human comparison, which failed to reach significance despite a strong shift in average ΔTPM, likely because only nine conserved TE-derived loops passing both distance filters were present in the data set.

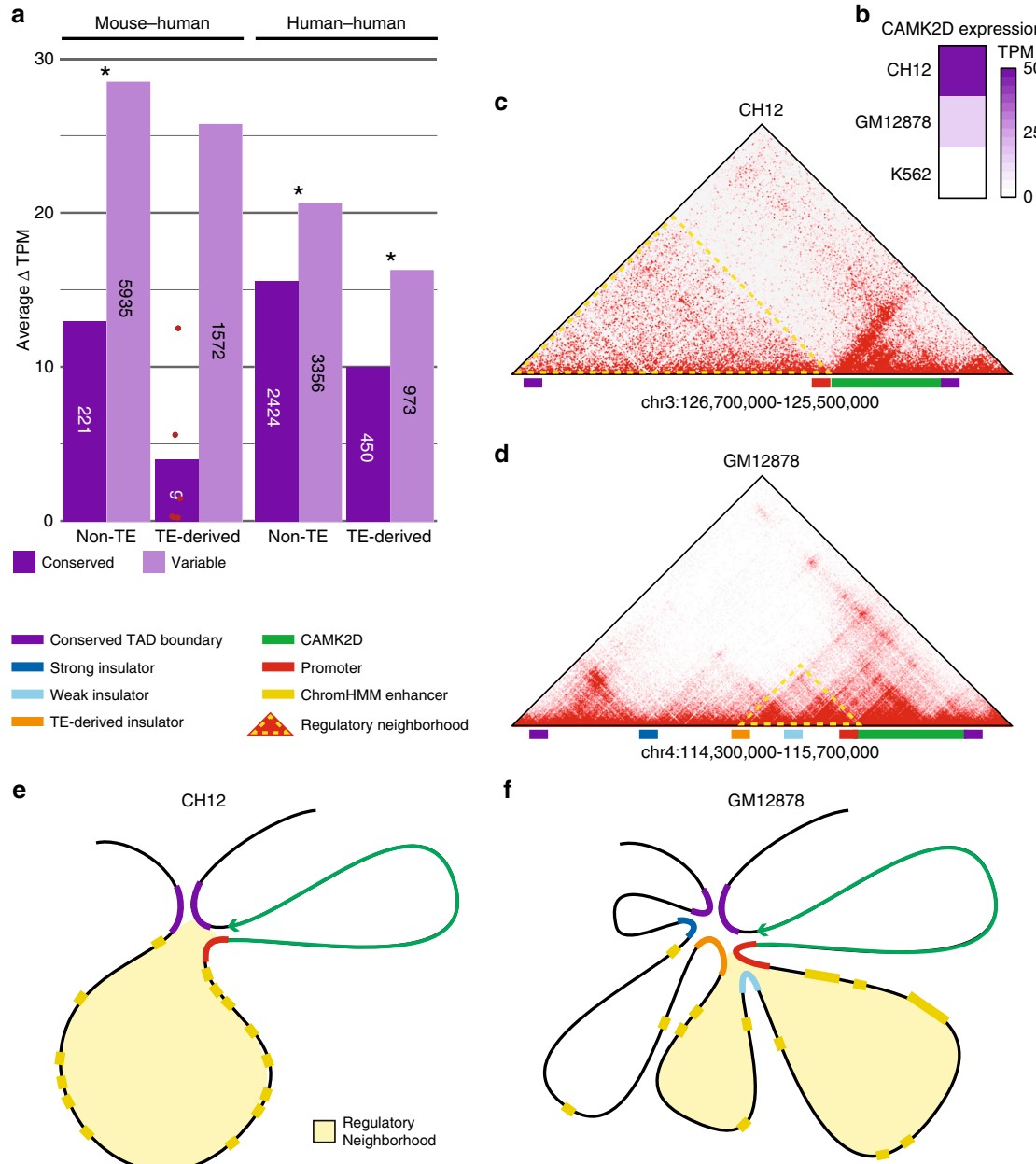

**Fig. 7 TE-derived and native variable loops are associated with variable gene expression. a** Bar plots show average differences in target gene expression, expressed as Δ*TPM* (see "Methods" for derivation), across species and cell types for conserved and variable enhancer–promoter loops. Loops are defined as "TE-derived", if one or both of the CTCF sites defining the loop boundaries is embedded within a TE and "native" otherwise. Bars are annotated with the number of contributing observations, with red points indicating observed Δ*TPM* values for conserved, TE-derived loops in the mouse–human comparison, for which only nine observations were available. *One-sided Wilcoxon *p*-value < = 0.01. **b–f** A variable enhancer–promoter loop associated with variable *CAMK2D* gene expression. **b** CH12, GM12878, and K562 *CAMK2D* expression in transcripts per million (TPM). **c** Hi-C plot for the TAD enclosing *CAMK2D* in mouse CH12 cells. Relevant features are highlighted along the *x*-axis (see color key). The *CAMK2D*-regulatory neighborhood (outlined in yellow) extends from the promoter to the distal TAD boundary, and is enriched throughout for enhancer–promoter interactions. **d** Hi-C plot showing the *CAMK2D* TAD in human GM12878 cells. Three CTCF-tethered loop anchors appear to act as insulators, dividing the TAD into four distinct subloops. A TE-derived CTCF-binding site insulates the *CAMK2D* promoter from contact with the two furthest-distal loops, thus restricting the *CAMK2D*-regulatory neighborhood to two proximal subloops separated by an embedded, native insulator element. Both of these subloops are enriched for enhancer–promoter interactions.
**e**, **f** Illustration of chromatin loops detected in mouse CH12 (**e**) and human GM12878 cells (**f**). Relevant features in both genomes are indicated by colored bars (see color key), and *CAMK2D*-regulatory neighborhoods are highlighted with light yellow fill. **e** In CH12 cells, multiple enhancers are free to interact with the *CAMK2D* promoter, evident as diffuse enrichment within the regulatory triangle in the CH12 Hi-C map (**c**). **f** In GM12878 cells, the *CAMK2D*-regulatory neighborhood is much more restricted. The 5′ boundary is demarcated by a TE-derived CTCF-binding site, which sequesters further-distal enhancers falling within two distal loops, included in the CH12 *CAMK2D*-regulatory neighborhood, from promoter interactions. The two proximal loops remaining in the regulatory neighborhood contain both multiple enhancer elements (Supplementary Fig. 13H) and an internal insulator element possibly refining enhancer–promoter interactions between loci in both loops.

To further explore the mechanisms by which TE-derived looping variability may elicit gene expression changes, we selected a variable loop within the TAD enclosing *CAMK2D* for further study. *CAMK2D* is the only expressed gene within a conserved TAD present in all three cell types (Fig. 7b–d; Supplementary Fig. 13D–F). RNA-seq data show variable expression ranging from high expression in CH12 to extremely low expression in K562 (Fig. 7b). Hi-C maps for these cell types show notable differences in looping that affect enhancer–promoter interactions within this TAD, corresponding well with observed differences in *CAMK2D* expression (Fig. 7b–f; Supplementary Fig. 13D–H). The strongest level of expression corresponds to a relatively large regulatory neighborhood containing several annotated enhancers, all of which appear to interact with the *CAMK2D* promoter in CH12 cells, leading to robust gene expression (Fig. 7c, e, Supplementary Fig. 13D, G). In comparison, while the orthologous region of GM12878 contains a similar number of enhancer annotations, the TAD is divided by three embedded insulator elements, one of which is anchored at a TE-derived CTCF-binding site (Fig. 7d, f; Supplementary Fig. 13E, H). This site defines the 5′ boundary of the *CAMK2D*-regulatory neighborhood, insulating the promoter from contact with the two furthest-distal intra-TAD loops, precluding a subset of possible enhancer–promoter interactions. The regulatory neighborhood is further divided into two subloops, both of which are enriched for enhancer–promoter and enhancer–enhancer interactions. This bipartite structure may add flexibility to *CAMK2D* regulation by separating enhancers into distinct groups that can be switched on and off independently from each other, the net result being intermediate baseline expression. In K562, by contrast, no enhancer–promoter interactions are evident within the *CAMK2D* TAD (Supplementary Fig. 13F), the entire TAD is depleted for active chromatin states (Supplementary Fig. 13H) and no expression is detectable in RNA-seq experiments (Fig. 7b). These results suggest that one mechanism by which TE-derived CTCF-binding sites produce variable gene expression is creation of tissue-specific insulator elements that refine regulatory neighborhoods in specific cell types. This mechanism appears to be distinct from the recently demonstrated association between variable gene expression and TE-derived dynamic TAD boundaries[40], demonstrating that smaller-scale looping changes may also directly lead to variable gene regulation.

## Discussion

Here we present evidence that even though transposable elements are typically considered deleterious and are depleted from most types of functional sequence, they have broadly impacted CTCF binding and contribute strongly to divergent chromatin looping in human and mouse with concomitant effects on variable gene expression. TE amplification has shaped the 3D genomic landscape throughout mammalian evolution by distributing novel CTCF-binding sites throughout host genomes. Under the right conditions, these may divert CTCF from existing binding sites and create novel chromatin loops that in turn influence gene regulation. Although most studies have focused on the impact of CTCF-enriched TE types to the genome, we find evidence that all TE types with an embedded CTCF motif may affect chromatin looping, and that the genomic magnitude of their effects reflects their abundance and the strength of the CTCF motif they carry. We present direct evidence to contradict the longstanding assumption that TE activity has had little effect on CTCF binding within primates. On the contrary, we demonstrate that TE activity has contributed ~1/3 of active CTCF-binding sites in both the human and mouse genomes, while CTCF-enriched TE types represent only a minority of TE-derived CTCF-binding sites in

the human genome. Furthermore, estimated age distributions for many enriched TE types predate rodent–primate divergence by a significant margin, suggesting that TE expansions have contributed novel CTCF-binding sites throughout mammalian evolution. Our results show that these sites often create looping variations with the potential to alter TADs and gene regulatory neighborhoods.

It seems likely that amplification of TE types containing a strong CTCF motif may divert CTCF binding from nearby existing sites with weaker motifs, thus altering loop formation within their host genome. The results of two recent studies[40,41] support this hypothesis by demonstrating experimentally that gain of TE-derived CTCF-binding sites or subsequent loss of CTCF binding at a nearby site can directly lead to formation of alternate chromatin loops. We demonstrate that TE-derived and native CTCF-binding sites are functionally equivalent as loop anchors, thus their net effect on genome structure and function likely scales with their distance from the existing CTCF sites they supplant. If sufficiently close together, the biological effects of a TE-driven CTCF insertion may be negligible, as exemplified by recently described conserved chromatin loops anchored by species-specific TE-derived CTCF sites[41]. While the net effect of these events may be selectively neutral, the functional redundancy these sites introduce may provide an evolutionary "safety net" that aids in maintaining higher-order chromatin structure. Indeed, these sites may actually accelerate CTCF-binding site turnover by decreasing selective pressure on neighboring sites while still having relatively little effect overall chromatin structure. At larger distance scales, the functional consequences of novel loops likely depend on the magnitude of any effects they have on TAD boundary formation. For example, TAD variability that causes differential inclusion of genes is likely to produce differential gene expression. This was recently confirmed in a study demonstrating that HERV-H elements introduce dynamic TAD boundaries in differentiating human cardiomyocytes, and these variable TADs contribute directly to expression changes in key cardiac genes[40]. We further show that smaller-scale changes in looping, at the sub-TAD level, may generate observable effects on gene expression by altering the regulatory neighborhoods of their target genes. These may elicit differential gene expression by refining enhancer–promoter contacts, thus changing how genes are used across cell types and species, or how they are regulated in response to environmental stimuli.

Throughout this study, we saw a strong link between TEs and variability. Although many TE-derived CTCF-binding sites are conserved in their placement between human and mouse, they contribute disproportionately to differential CTCF binding (Fig. 2a) and variable looping (Fig. 6a–c) across cells and species. This may reflect an underlying tendency toward selective neutrality for TEs that remain in the genome (i.e., those that are not selectively eliminated due to immediate lethality or long-term deleterious effects), or alternately there may be some selective advantage to having a large pool of preferred and alternate CTCF-binding sites from which to form chromatin loops. This may trace back to the stabilizing effects demonstrated by Choudhary et al.[41]. However, we demonstrate that TEs make proportionally larger contributions to variable loops, suggesting that the ability to readily adopt alternative chromatin conformations may itself be necessary and perhaps selectively advantageous. For example, recent results suggest that looping variability is necessary for TAD formation and chromatin structural dynamics. Contrary to the original description of TADs as stable and highly conserved structures[3,4], TAD boundaries appear to be determined stochastically through random interactions between multiple preferred and alternate loop anchors, with pronounced variability across developmental stages, tissues, and individual cells[22,57]. Computer

simulations show that chromatin loop extrusion, currently the most prevalent proposed mechanism for loop formation, requires looping variability in order to recapitulate observed chromatin contact maps[31,49,58]. In accordance with these predictions, our results show that individual loop anchors rarely interact with only one downstream partner and that TE activity is a primary source of this variability. Therefore, while TEs may inherently be selfish sequences, their continued activity may serve to maintain a sufficient pool of CTCF-binding variability to support proper chromatin structural dynamics. Thus, even though many variable chromatin loops may not have individually identifiable biological effects, they may contribute to the ability of the host genome to switch between various 3D conformations. This may further enable rapid gene expression changes such as those accompanying differentiation and response to environmental stressors.

We also note that environmental stressors can directly trigger TE activation and expansion[59]. While it seems likely that such activity is opportunistic from the TE's point of view, it may have the side effect of increasing regulatory flexibility within the host genome at key points in evolution. This may explain why several of the TE expansions that have distributed CTCF-binding sites in the human and mouse genomes coincide closely with estimated divergence dates[30,53]. In such a model, exapted CTCF-binding sites may contribute to adaptability by increasing gene regulatory diversity within the population. While many insertions would lead to immediate lethality or deleterious effects, some would facilitate gene regulatory changes that increase fitness, allowing certain members of the population to adapt to changing selective pressures. Furthermore, many TE-derived loop anchors from such events, as well as those originating from temporally diffuse amplifications, may be maintained in the genome as selectively neutral variation that is, over time, coopted as a source of regulatory variability. The broad range of TE age distributions we observed relative to the estimated human–mouse divergence date (Fig. 3a) suggests that both positive selection on novel TE insertions and cooption of evolutionarily neutral TE-derived CTCF sites through genetic drift have likely contributed to the TE-derived looping variability we observe. However, our current methods cannot discern between these two distinct evolutionary modes, and this remains an area of active speculation.

We were surprised by the number of TE types carrying CTCF-binding sites within their consensus sequences. This suggests that CTCF may serve important roles in TE biology, perhaps facilitating replication, integration into the host genome, or otherwise harnessing host genomic mechanisms to enhance their long-term survival. It is possible that these TE types rely on chromatin looping to gain access to the host's transcriptional machinery or utilize CTCF-associated double-strand breaks[60] to facilitate integration into the host genome. Alternately, CTCF-binding sites may serve as nucleation sites for epigenetic modifications that allow TEs to evade host surveillance, thus enhancing their survival. In any case, the embedded CTCF-binding site may give these TE types a competitive advantage compared to TEs that lack CTCF, thus increasing their long-term survival. However, while TE sequences may be inherently selfish, beyond simply being tolerated in their host genomes, they are often repurposed and used to fulfill necessary host processes. This suggests that they may actually exist as genomic symbionts rather than parasites.

In planning this analysis, we had to decide which cell types to include from a range of published data sets. Although CTCF ChIP-seq and 3D chromatin contact data sets are available for many human cell types, our choices were limited by the availability of mouse data sets for closely matched cell types. In the end, we chose to focus on ENCODE tier 1 cell lines because of the richness of available data sets. It is important to note that many of these cell lines have karyotypic abnormalities, including fusions,

amplifications, segmental duplications, inversions, and deletions, which may be relevant to intrachromosomal looping. Among the three cell types used in our cross-cell looping comparisons, K562 has the most abnormal karyotype. The primary difference between K562 cells and normal cells is that K562 are near triploid. Importantly, this should not affect the intrachromosomal interactions observed in this study. While K562 cells also contain several rearrangements, their scale far exceeds the size of most loops observed in this data set[61] and their overall effects on genomic structure are minimal. Furthermore, the number of loops observed in K562 exceeds the number of rearrangements by nearly an order of magnitude. Thus, we expect very few loops to span rearrangement breakpoints; these should have minimal impact on our estimates of conservation. As evidence of this, we point out the close agreement we observed in conservation levels in comparisons of CH12 to GM12878 and K562. While CH12 and K562 showed a nominal decrease in conservation relative to GM12878 (Supplementary Table 6), this likely reflects biological differences between the lymphoid and myeloid lineages rather than karyotypic differences. In fact, results from CH12 and K562 were in close agreement with GM12878 in all looping comparisons (Supplementary Fig. 6, Supplementary Fig. 11). While broadening the scope of this comparison across more cell types would certainly deepen our understanding of chromatin looping evolution, this would require substantial investment in generating mouse ChIA-PET and Hi-C data sets.

In conclusion, we speculate that TE-driven CTCF-binding expansions have contributed to regulatory flexibility throughout mammalian evolution by expanding the number of chromatin loop anchors within their host genomes. Our results complement those of Choudhary et al.[41] showing that TEs make important contributions to cell-specific and species-specific loops in addition to their possible functions in stabilizing conserved chromatin structure. These variable loops may serve necessary functions in 3D chromatin dynamics as well as increasing the number of alternate chromatin conformations possible across cell types, perhaps contributing to adaptability. We show that TE-induced looping variability is a major contributor to differential gene expression, demonstrating that sub-TAD-scale looping variations may alter gene expression by refining cell-specific enhancer–promoter interactions, thus extending recent observations that TE-derived TAD boundaries can directly alter gene expression[40]. We postulate that TE-induced population-level looping variability in the MRCA of human and mouse may have conferred adaptive advantages that allowed certain individuals to flourish in the face of changing selective pressures. This, in turn, may have led to divergence between subpopulations as they adapted to distinct evolutionary niches, eventually leading to speciation. This work advances our understanding of the relationship between TEs and their host genomes, raising important questions about the interplay between the role of CTCF in TE biology, the necessity of CTCF variability in host chromatin dynamics, the evolutionary forces driving looping variability, and their effects on adaptation to a changing environment.

## Methods

**Overlap of CTCF occupancy between human and mouse.** For step 1, CTCF-binding sites for human GM12878 and K562, and mouse CH12 and MEL cells were retrieved from the ENCODE repository, using all released data sets (Source Data File). Biological and technical replicates were combined using bedtools merge[62] and stored in narrowPeak format. For merged peaks, we assigned the summit location as the centroid of the peak summits for all constituent binding sites. For broadPeak records, the midpoint of the binding site was used as a proxy for the narrowPeak summit. This procedure was repeated in pairs of cell-wise files from the same species in order to determine the union set of CTCF-bound sites in each species.

The next step involved comparison of species-wise sets of CTCF-binding sites to determine the extent of overlap between human and mouse CTCF-binding

landscapes. Prior to cross-species mapping, a unique identifier was assigned to the name column of the input files to facilitate backward comparisons of mapped features across species. CTCF-binding peaks were then mapped across species using a modified version of bnMapper[63], with an added option to retain peaks spanning multiple chains by keeping the longest subalignment, following the convention used by the liftOver utility[64]. This modified version is freely available at https://github.com/Boyle-Lab/bx-python.

Step 3 involved merging native CTCF-binding peak locations and CTCF-binding peaks mapped from the other species into union sets representing the locations of all mappable CTCF-binding sites across species. Comparisons between the merged narrowPeak files prepared in step 2 were made with bedtools intersect[62] in order to apply labels, indicating the specie(s) in which each site was occupied.

The sets resulting from step 3 were then intersected with the transposable element locations from RepeatMasker annotations[50], excluding "Low_complexity", "Satellite", "Simple_repeat", "tRNA", "rRNA", "scRNA", "snRNA", and "srpRNA" families. Bedtools intersect was used to identify all CTCF sites, in which a transposable element was detected within $+-50$ bp of the ChIP-seq peak summit, and an additional column of labels was added accordingly.

In step 5, procedures from steps 3 and 4 were applied to the species-wise unmapped CTCF ChIP-seq peaks. These were loaded into R data frames along with the union sets produced in step 5, and unique identifiers applied in step 2 were used to identify sites that did not cross-map with bnMapper. These were appended to human- and mouse-referenced union sets to yield complete sets of all known CTCF-bound sites across both genomes. The contribution of TEs to shared and species-specific binding sites was visualized using the VennDiagram R library. Sizes and shapes of individual plot segments were adjusted manually to approximate their proportional contribution to the union data sets (Fig. 1a).

The expected numbers of TE-derived human-specific and mouse-specific binding sites were calculated based on overlaps between species-specific CTCF-binding sites and randomly selected windows following the size distribution of TE-derived CTCF-binding sites in each species. We selected $N$ random background regions from the given genome, where $N$ is the number of species-specific CTCF sites derived from TEs. We then counted the number of overlaps between background regions and species-specific CTCF-binding sites. We used the median number of overlaps observed over 1000 random trials as the expected number of TE/CTCF overlaps.

**CTCF-binding site enrichment in human and mouse transposons**. We first sought to identify transposable element families that may have distributed CTCF-binding sites in humans and/or mice. This inquiry extends the findings presented in three landmark studies, in which enrichments of transcription factor binding sites, including CTCF, were identified in several TE families in humans and mice[32,33,65]. Intriguingly, Schmidt et al., the only study to look specifically for enrichment of CTCF binding sites in primate cells, failed to find any significant enrichments in human, despite strong enrichments in mammalian species stretching back to opossum. We wondered whether the reliance of their methods on enrichment of species-specific k-mers at CTCF-binding sites influenced their findings. We tested for enrichments using two approaches: binomial tests based on methods used in Bourque et al.[33], and permutation tests based on methods presented in Chuong et al.[65]. In both methods, we rely solely on the observed frequency of CTCF binding within each TE type compared with a random expectation to determine enrichments. We performed both analyses on genome-wide sets of CTCF-binding sites, and on a more restricted set of CTCF sites located in mouse and human cis-regulatory modules (CRMs) previously reported by us[66].

Merged CTCF ChIP-seq peak files were loaded into a Hadoop Hive database after adding unique id, species, cell, and target columns. The locations of all annotated transposable elements for human and mouse were retrieved from RepeatMasker annotations[50] (Source Data File). Data were converted to bed format and all records, excluding "Low_complexity", "Satellite", "Simple_repeat", "tRNA", "rRNA", "scRNA", "snRNA", and "srpRNA" families. These were annotated with a unique id, species, and the name and distance to the transcription start site of the nearest gene according to bedtools closest[62] and the knownGenes table for hg19 or mm9 genomes[67]. These were loaded into the database and intersections with CTCF-binding sites were identified with a series of hive queries. Intersections were based on the CTCF ChIP-seq summit location, which was required to fall within the boundaries of a TE annotation. The resulting data were output as a tab-delimited text for further processing in R.

We tested for enrichment of CTCF-binding sites within individual TE types using three methods: individual binomial tests using the average genome-wide rate of CTCF-binding within TEs as the expected binomial frequency; individual binomial tests using binomial expected frequencies based on CTCF binding within each TE type in permuted data; and enrichment tests based on empirical cumulative density functions computed from the permuted data sets.

For binomial tests (Supplementary Table 1), we calculated the genome-wide fraction of TEs containing CTCF-binding sites within all four cell types separately and used these as the binomial expected frequencies. Within each cell type, we performed individual binomial tests for every TE type, in which CTCF binding was observed and adjusted $p$-values for multiple testing using the Bonferroni method. We applied three criteria for significant enrichment within a TE type: $p$-value $<=$

$1 \times 10^{-4}$, at least 25 CTCF-bound TE insertions, and a CTCF-binding rate of at least 1% within the given TE type.

For permutation tests (Supplementary Table 2), we used a method originally reported by Chuong et al.[65]. Starting with the R data frames used in binomial tests, we performed 10,000 random permutations of the CTCF-TE data by shuffling the associated TE types. For each permutation, a Fisher–Yeats shuffle was performed on the name column of the whole-genome repeatMasker annotations using fyshuffle (fgpt R package). In order to maintain the insertion biases of each TE type, shuffling was performed separately within six distance-based bins relative to the nearest transcription start site for each TE insertion. Shuffled names were then applied to the corresponding records for CTCF-bound repeats. The number of times a given TE type was observed among records originally labeled with that family was recorded at each permutation, and resulting counts were used to generate an empirical CDF for each TE family using the ecdf function. Empirical $p$-values for enrichment of CTCF sites in each TE type were computed by plugging the observed number of CTCF-binding events into the corresponding eCDF functions, and a Bonferroni multiple testing correction was applied. We applied the same set of criteria used in our binomial tests to assess significance among these results.

**Assessing motif–word frequencies in enriched transposons**. We performed our motif–word frequency enrichment analysis by replicating the procedures presented in ref. [32] and applying them to our own CTCF motif predictions within human and mouse CTCF-bound repeats. Initially, CTCF-bound repeat insertions in the human and mouse genomes were identified by intersecting repeatMasker annotations, excluding "Low_complexity", "Satellite", "Simple_repeat", "tRNA", "rRNA", "scRNA", "snRNA", and "srpRNA" families, with ChIP-seq peak summit locations using bedtools intersect[62], and fasta sequences were extracted from the hg19 and mm9 genomes with bedtools getfasta[62]. FIMO motif prediction was performed using a previously published CTCF position weight matrix[68], using default parameters and a maximum site count of 1,000,000. Predictions were converted to bed format, retaining the sequence of the predicted binding sites as the final bed field. These were read into R data frames and motif-words, distinct 20-mers contributing to the pools of CTCF-binding site predictions, were enumerated within human and mouse. Observed bound word counts, denoted $occ_{i,j}$, were converted into normalized word counts, $nocc_{i,j}$, by applying a scale factor (Eq. (1)).

Equation 1: Calculation of normalized word counts. $i$ = motif–word number, $j$ = species, $L$ = motif length (base pairs).

$$nocc_{i,j} = occ_{i,j} \left( \frac{\sum_{i=1}^{n} occ_{i,j}(L)}{1,000,000} \right)^{-1}$$

Odds ratios denoting species-specificity were then calculated (Eq. (2)) and used to assess species-specificity. As in Schmidt et al.[32], we considered all motif-words with a normalized occurrence rate of at least 8 and an absolute odds ratio of 2 or greater as species-specific.

Equation 2: Odds ratio calculation for assessment of motif–word species-specificity. $nocc_i(\text{hg})$ = human normalized word count for gene $i$. $nocc_i(\text{mm})$ = mouse normalized word count for gene $i$.

$$\ln \frac{nocc_i(\text{hg})}{nocc_i(\text{mm})}$$

We next tested for association of individual human and mouse TE types with species-specific motif-words by comparing their occurrence rates within CTCF-bound TE elements of a given type to their occurrence rate in the rest of the genome. We first isolated CTCF-bound motif-words within the human and mouse genomes by intersecting the previously prepared genome-wide CTCF motif predictions, with 100 bp windows surrounding CTCF ChIP-seq peak summit locations in each species using bedtools intersect[62]. These were read into an R data frame. For each TE type observed in the CTCF-bound repeat data, individual Fisher's exact tests were performed, comparing the observed number of species-specific versus shared motif-words in bound repeats compared with background sequences, defined as all CTCF-bound sequences in the given genome excluding those of the given repeat type. $P$-values were adjusted by applying a Bonferroni correction, and a one-sided $p$-value threshold of $1 \times 10^{-40}$ was used to determine significance, as in Schmidt et al.

**Age estimation for CTCF-associated transposable elements**. We estimated the ages of TE insertions by dividing the percent divergence from the consensus sequence for each record (reported in the RepeatMasker annotations), by an estimate of the mutation rate per base per year for each species. Although there is some disagreement in the community about actual mutation rates for human and mouse, we chose to use estimates presented by Kumar and Subramian[69]. For human, we used the consensus mammalian rate of $2.2 \times 10^{-9}$ substitutions/base/year, which agrees with the widely-accepted rate presented by the Human Genome Sequencing Consortium[70]. For mouse, we used a rate of $2.4 \times 10^{-9}$, obtained by dividing the human rate by 0.091, to account for the 9.1% faster mutation rate in rodents relative humans reported by Kumar and Subramian[69]. This rate is substantially slower than the rate of $4.5 \times 10^{-9}$ reported by the Mouse Genome Consortium[71], which was used to prepare age estimates in Schmidt et al.[32]. However, Kumar and Subramian make a compelling argument that biased

substitution patterns can artificially inflate estimated mutation rates and, thus, we opted to use the slower rate as it accounts for these effects. Boxplots were prepared in R using ggplot2 to produce Figs. 3a and 6g. Because the TE consensus sequence represents the average conformation rather than the true ancestral form, this method produces inexact estimates for insertion such that the ages of individual TE copies may be over or underestimated. However, this should not significantly skew the overall age distributions.

**Phylogenetic gain and loss analysis of CTCF-binding sites.** In order to assign labels denoting evolutionary history to each CTCF-binding site present in the human and mouse genomes, we developed a method, mapGL.[55] We obtained liftover chains for human (hg19) to mouse (mm9), and for human and mouse to three outgroup species: dog (canFam2), horse (equCab2), and elephant (loxAfr3), from the UCSC Genome Browser download portal[72]. We then constructed reciprocal-best alignment chains, representing one-to-one relationships of syntenic blocks between each genome, following standard procedures: [http://genomewiki. ucsc.edu/index.php/HowTo:_Syntenic_Net_or_Reciprocal_Best].

We ran mapGL on human and mouse inputs separately, using the reciprocal-best human-to-mouse chain to map human elements, and the reversed human-to-mouse chain to map mouse to human. Relationships between the target and query species and outgroup species are described by the Newick tree: (((hg19,mm9), (canFam2,equCab2)),loxAfr3). mapGL.py was invoked with the "–input_format narrowPeak" option, to include the mapped location of narrowPeak summits in output whenever possible.

We first intersected 50 bp windows surrounding each human and mouse ChIP-seq peak summit with repeatMasker repeats, as described in the "CTCF-binding site enrichment in human and mouse transposons" section. Phylogenetic labels were then applied with mapGL, and the results were annotated with additional data from the original repeatMasker files. These were further analyzed in R. Specifically, the contribution of each phylogenetic class to each CTCF-enriched repeat type was assessed by plotting the fraction of elements within each type assigned as orthologous, gained, or lost on a given branch using the ggplot2 R package.

**Motif scores within CTCF-enriched ancestral repeats.** We first computed log-odds scores for matches to a previously published CTCF position weight matrix (PWM)[68] at every position in the human and mouse genomes using a custom Perl script (score_motifs.pl [https://github.com/adadiehl/score_motifs]). These were stored in wig format and converted to bigWig files using the wigToBigWig utility[64]. We made use of the rtracklayer package to retrieve motif scores from these bigWig files for TE instances annotated as orthologous across human and mouse. For each of these regions, scores were retrieved for a 50 bp window centered around the summit of the embedded CTCF ChIP-seq peak. This was repeated for the orthologous location in human or mouse, and the maximum scores were stored for both species. To obtain the CTCF motif score distribution within consensus elements for human-only TEs, we first extracted fasta format consensus sequences for each TE type from Repbase version 23.09[52] using a purpose-built tool (extractAncestral.py 0.0.1 [https://github.com/adadiehl/repeatMaskerUtils]). Each consensus sequence was scored using score_motifs.pl [https://github.com/adadiehl/score_motifs] and the same CTCF PWM used to generate the bigWig files, and the maximum scores from each were retained. Score distributions were visualized using the ggplot2 R package. We assessed the significance of the observed difference between human and mouse score distributions using a Wilcoxon signed-rank test, and between consensus, human, and mouse using Wilcoxon rank-sum tests.

**MANGO analysis of RAD21 ChIA-PET data.** We retrieved ChIA-PET data for human GM12878 and K562 from the ENCODE download portal in fastq format (Source Data File). After careful evaluation of FastQC[73] results on each input file, we elected to proceed without adapter trimming. Paired reads in fastq format were aligned separately with BWA mem[74] with default parameters. Mapping quality was evaluated with SAMtools flagstat[75] found to be > = 94% for all but two files. Samtools view was then used to filter out unmapped and secondary reads (SAM flags 4 and 256), and those with quality scores less than 30. Filtered reads were then sorted by *X* and *Y* coordinates, and a custom script was used to assemble paired reads into bedpe format. Bedpe files for all biological and technical replicates were then concatenated and processed with the MANGO pipeline, starting with stage 3.

**Contribution of transposons to human and mouse loop anchors.** RAD21 ChIA-PET loops for human GM12878 and K562 cells, and Hi-C loops for the same human cells and mouse CH12 cells, were first filtered to include only loops containing a CTCF ChIP-seq peak at both anchors. If multiple CTCF ChIP-seq peaks overlapped a loop anchor, we kept only the peak with the strongest signal value. Loop anchor coordinates were then trimmed to the boundaries of their respective overlapping ChIP-seq peaks. The location of the overlapping ChIP-seq peak summit was included in the record as an additional field. The trimmed and filtered loop loci were then read into a data table in Hadoop hive database table. We then intersected these loops with the CTCF-TE associations prepared in our analysis of CTCF-enriched TEs by comparing the CTCF peak summit locations.

ChIA-PET and Hi-C loops were separately intersected with CTCF motif predictions. CTCF motif predictions were prepared in-house using a custom script.

We used a previously published CTCF position weight matrix[68] and calculated simple log-odds scores relative to the equilibrium nucleotide frequencies for each base for overlapping, 20-bp windows spanning the human and mouse genomes. In order to maximize the fraction of CTCF-bound sequences to which motifs could be assigned, we retained all predictions with log-odds scores exceeding 0, although nearly 70% of CTCF-bound sites contained a motif with a log-odds score > = 10. We associated CTCF motifs with RAD21 ChIA-PET and Hi-C loops by extending a 50-bp window surrounding the CTCF ChIP-seq peak summit. We then used an R script to assemble a data frame for each cell type, containing a row for each RAD21 ChIA-PET or Hi-C loop, and columns indicating TE presence and CTCF motif presence and orientation in the right and left loop anchors. These data frames were used to collect counts presented in Fig. 4, Supplementary Fig. 3, and Supplementary Table 5.

**Contribution of transposons to conserved and variable loops.** In order to categorize loops based on sharing between cells, we devised a simple classification scheme based on physical overlap of loop anchors. Loops in the query species were classified by looking for overlaps between their left and right anchors and loop anchors in a set of target loops from another cell. If the left and right anchors both mapped to anchors from the same loop in the target set, a loop was assigned to class C—fully conserved. Class B2 designated loops where both query anchors overlapped target anchors, but target anchors were from different loops. Class B1 designated loops where only one query anchor overlaps a target anchor, and B0 designates loops where neither query anchor overlaps a target anchor. To accommodate cross-species comparisons, we added classes N0, N1A, and N1B, which represent anchors where one or both loop anchors are present in non-orthologous sequence. N0 denotes loops where both anchors are non-orthologous to the target species. N1A denotes sequences where one query anchor is both orthologous and overlaps a target loop anchor. N1B denotes loops where one query anchor is orthologous but does not overlap a target loop anchor. Loops from all three cell types for which we have loop data (GM12878, K562, and CH12) were assigned to these classes using another adaptation of bnMapper, which we call mapLoopLoci. This tool is available from our github repository [https://github. com/adadiehl/mapLoopLoci]. We assigned conservation classes to loops for all pairwise combinations of cells and combined the results into an R data frame. We then intersected these data with the results from our previous analysis of TE–loop intersections based on previously assigned loop ID numbers and counted the fractions of loops in each conservation class contributed by TEs. In the case of species-specific and cell-specific loops, we required that the TE insertion overlap the loop anchor(s) unique to the query cell in order to count toward the total number of TE-derived loops.

**Correlation of conservation with loop strength and TE load.** To determine whether any correlation exists between loop strength, loop conservation, and TE content, we compiled PET counts for RAD21 ChIA-PET loops in human GM12878 and K562 cells, and observed Hi-C contact counts for mouse CH12 cells, in each of the seven conservation classes defined in "Contribution of transposable elements to conserved and species-specific chromatin loops." Boxplots for TE-derived and non-TE-derived fractions for each set of scores were produced using the ggplot2 R package in order to visualize any score trends. This process was repeated for all pairwise combinations of cells. Wilcoxon rank-sum tests were performed to determine the significance of any trends toward higher or lower scores for all pairs of conservation classes within each comparison, and between TE-derived and native loops within each conservation class across all comparisons. Resulting *p*-values were then plotted as heatmaps with the ggplot2 R package.

**Ages of TE elements in each loop conservation class.** To determine if any trends were evident between conservation classes and the estimated ages of TEs contributing to each class, we estimated the ages for all TE insertions within each conservation class in mouse-to-human and human-to-mouse comparisons according to procedures reported in the "Age estimation for CTCF-associated transposable elements" Methods section. Score distributions for each conservation class were rendered as boxplots and visually compared to identify any notable trends. In order to test for a significant linear correlation between conservation classes and estimated TE ages, we assigned numeric values to each conservation class and fitted a linear model relating conservation class and estimated TE age with the "lm" function in R. Goodness-of-fit and significance of the observed linear trend were determined by applying the summary function to the fitted lm model. We further tested for significant differences in estimated TE age distributions between pairs of conservation classes and TE age in mouse–human comparisons using individual Wilcoxon rank-sum tests.

**Contribution of enriched TEs across conservation classes.** To determine if there is a relationship between loop conservation and CTCF-enriched TE types, we labeled each TE insertion in the loop conservation data set as mouse-enriched, human-enriched, shared, or non-enriched. The contribution of each enrichment category to loops in each conservation class was visualized by iteratively applying the "table" function in R and plotting the resulting table of counts as stacked bar graphs with ggplot2. For the mouse–human comparison (Fig. 6d), the observed

counts for human-enriched and shared TE types were combined for clarity. This only affected the "C" and "B2" conservation classes.

**Correlation of loop conservation and evolutionary constraint**. We first retrieved phastCons 46-way placental mammal conservation scores in wig format from the UCSC download portal[67]. These were subsequently converted to bigwig format with the wigToBigWig utility[64]. We used an R function, making use of the rtracklayer Bioconductor package, to retrieve phastCons scores for 500-bp windows surrounding the annotated CTCF peak summit location within all TE-borne loop anchors, and the resulting matrix of scores was summarized with the colMeans function. This process was applied to all loops in each conservation class, and mean score vectors were stored in a data frame and plotted as line graphs with ggplot2. Because PhastCons scores are calculated using a much deeper and broader phylogeny than was used in our analysis, they allowed us to assess functional constraint at many loci where human and mouse lack sequence orthologs. Furthermore, the plots represent an average over all sites in each class, thus mitigating the effects of sequences lacking sequence orthologs in any species.

**Overlap with topologically associating domains**. We obtained published predicted locations of topologically associating domains (TADs) for CH12, GM12878, and K562 cells from the GEO repository[3] (Table 1). We converted the primary data from the Arrowhead format to a BED format where the start and end coordinates correspond to the X and Y coordinates defining the TAD boundaries within the Hi-C contact matrix and read into R for further processing. In order to identify loops in our data set that correspond to known TADs, we first found the intersection with our loop data set using the findOverlaps method from the GenomicRanges R library, with default options. This identified which loops overlapped known TADs, but did not indicate which loops correspond to entire TAD regions. To do so, we defined upstream and downstream boundary regions for each TAD by extending a window + −10 kb around their X and Y coordinates. We then looked for loops where the upstream and downstream anchors overlapped the upstream and downstream boundaries of a single TAD from the same cell type.

**Epigenetic properties of conserved and variable loops**. We obtained ChIP-seq data for CH12, GM12878, and K562 cells in bigWig format from the ENCODE download portal (Table 1). We used the rtracklayer package[76] to retrieve histone marks signals at each location in 20-kbp windows surrounding the annotated CTCF peak summits within loop anchors and stored them in data frame. For each histone mark and cell line, we compared the average signals in left and right portions of each 20-kbp window region and flip the direction if the right half has higher signal than left. Mean signal at each location was calculated using the colMeans function, and then divided by mean signals at randomly selected 20-kbp windows across the genome calculated following the same method. This process was applied to TE-derived and native loops anchors separately and processed score vectors were plotted as line graphs for each cell type using ggplot2. Then for human GM12878 and K562 cells, we filtered regions that are loops anchors in only one of the two cell types and followed the same steps to plot line graphs for these cell-type-specific loop anchors in each cell line.

**Analysis of target gene expression**. Initially, loop anchors for all loops in the data set were annotated with their maximal overlapping H3K4me3 and H3K4me1 signal using the same data sets from the "Epigenetic properties of conserved and species-specific chromatin loops" analysis, using the rtracklayer package[76]. These values were scaled from 0 to 1 within each sample and empirically determined thresholds were used to isolate putative enhancer–promoter, promoter–promoter, and enhancer–enhancer interactions by comparing the scaled values between the left and right loop anchors. The resulting sets of loops were visualized as heatmaps for presentation in Supplementary Fig. 13A.

All loop anchors within the data set were first annotated with their nearest genes and the distance to the associated transcription start site (TSS). We obtained gene annotations from the UCSC Genome Browser[64] knownGenes track for hg19 and mm9 assemblies and extracted the furthest-5′ TSS locations for all unique genes were extracted into BED files. We next annotated each loop anchor with the nearest gene by mapping the midpoint coordinates against the TSS BED files, using bedtools closest[62] with default options. The nearest gene and distance to the associated TSS were recorded in the R data frame. We made use of quantile-normalized RNA-seq expression data for CH12, GM12878, and K562 cells we previously prepared for another study[66]. Tabular gene expression data in transcripts-per-million (TPM) were read into an R data frame. Gene expression values were then associated with loop anchors using their nearest-gene annotations.

We estimated gene expression variation for each loop in pairwise loop conservation comparisons by intersecting the gene expression annotations for each loop with the loop conservation data set prepared in "Contribution of transposons to conserved and variable loops" and calculating ΔTPM for individual genes across pairs of query and target cells (Eq. (3)).

Equation 3: Calculation of gene expression variability. $TPM_i(q)$ = observed gene expression in query cell type. $TPM_i(t)$ = observed gene expression in target cell type.

$$\triangle TPM_i = |TPM_i(q) - TPM_i(t)|$$

Loops between gene promoters and distal sites were isolated by requiring one anchor within $<= 1$ kb from its nearest TSS and one anchor $>= 3$ kb from its nearest TSS. Average TPM values for genes adjacent to the promoter-proximal anchors of each loop were compared between conserved loops (conservation class C) and non-conserved loops (all other conservation classes) for TE-derived and non-TE-derived subsets in all possible pairwise combinations of cells (Supplementary Fig. 13B, C), and for data aggregated by species (Fig. 7a). Wilcoxon rank-sum tests (wilcox.test function in R with alternative="g") were used to assess the significance of trends toward larger values of TPM for target genes of variable loops compared to conserved loops.

**Reporting summary**. Further information on research design is available in the Nature Research Reporting Summary linked to this article.

## Data availability
The data that support this study are available from the corresponding author upon reasonable request. Data sets used for the analyses supporting the conclusions of this article are listed in the Source Data file, which also includes Supplementary Tables 2–6.

## Code availability
All scripts and code used to produce these results are available through github at https://github.com/Boyle-Lab/TE-Driven-CTCF-Loop-Evol. Custom scripts and software used in this analysis are available at the following repositories: bx-python 0.8.1 [https://github.com/Boyle-Lab/bx-python], MANGO Wolverine 1.1.9 [https://github.com/adadiehl/mango], mapGL.py 0.0.1 [https://github.com/adadiehl/mapGL], mapLoopLoci.py 0.0.1 [https://github.com/adadiehl/mapLoopLoci], extractAncestral.py 0.0.1 [https://github.com/adadiehl/repeatMaskerUtils], score_motifs.pl 0.3 [https://github.com/adadiehl/score_motifs].

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

## Acknowledgements

We would like to thank members of the Boyle lab and Dr. John Vincent Moran for critical reading and suggestions on the manuscript and analyses. This project was made possible through funding by the Alfred P. Sloan Foundation (FG-2015-65465) and National Science Foundation CAREER Award (DBI-1651614 to A.B.).

## Author contributions

A.G.D. and A.P.B. conceived and planned the study. All experiments and analyses were performed by A.G.D., with the exception of the analysis of epigenetic properties of conserved and species-specific chromatin loops, which was performed by N.O., A.G.D. prepared the manuscript with input from all authors. A.P.B. supervised the experiments, analysis, and data interpretation. All authors read and approved of the final paper.

## Competing interests

The authors declare no competing interests.
