## [Peer Review File · Nature Communications]

Reviewers' comments:

Reviewer #1 (Remarks to the Author):

The evolution of genome organization is essential to understand the functions of higher-order chromatin structures. Mounting evidence indicates that mobile elements are a major contributor to transcription factor binding, thus this manuscript is tackling an important transcriptional regulation question. Overall, the authors have conducted careful bioinformatic analysis and integrated multiple dimension data. My major concern is the content overlap with two related studies. One is from Zhang et. al. recently published in Nature Genetics (Transcriptionally active HERV-H retrotransposons demarcate topologically associating domains in human pluripotent stem cells). The other is from Choudhary et.al deposited in BioRxiv in Dec 2018 (Co-opted transposons help perpetuate conserved higher-order chromosomal structures). The authors need to clarify the novel findings in this manuscript that different from those two.

Another concern is that even though K562 cells have been widely used to identify many interesting functional genomic findings. It is not a good cell system for the comparison conducted in this manuscript. Many translocations, fusions and segment duplication events have been detected in the genome of K562 cells. Those mutations can lead to different levels of artifacts in chromatin interactions and thus make the human-mouse comparison difficult to interpret. With many recent published Hi-C, ChIA-PET and Hi-ChIP data, the authors have better options.

One weakness of this manuscript is that unlike the two studies mentioned, there are no experimental validation in this study. Thus, the biological meanings is relative weak.

Minor points:

- 1) The author used CTCF binding sites generated by the standard ENCODE pipeline. Since non-uniquely mapped reads are automatically filtered in this pipeline. A large proportion ME-derived binding can be lost.
- 2) Can the authors provide the definition of "latent CTCF"? (line 183)
- 3) The CTCF binding site gain and loss analysis were based on the whole peak regions. Due to the turn-over feature of TF, motif based analysis is more appropriate.
- 4) The hierarchy structure of TADs need to be considered when the author conduct the comparison.

Reviewer #2 (Remarks to the Author):

Review of "Transposable elements strongly contribute to cell-specific and species-specific looping diversity in mammalian genomes", NSCOMM-19-20964.

In this paper, the authors study the evolutionary origins of the DNA sequences defining the anchors of some 3D chromatin loops. There is a growing interest in 3D chromatin structures, which role in genome dynamics and gene regulation seems to be important, and the questions raised in the manuscript can now be addressed thanks to the development of various technologies and databases reporting 3D structure information. Here, the authors focused on chromatin loops associated with the protein CTCF, which is also known to affect the transcription rate of neighboring genes. CTCF binds specific sequences in DNA, and these binding sites are often located in Transposable Elements.

The authors' results confirm that about 15% of loop anchor sites can be located in transposable elements in human and mouse. The paper also shows that canonical TE sequences frequently contains sequences corresponding to CTCF binding sites in both species. Some of these sites are evolutionary conserved since the mouse-human divergence, suggesting some kind of conservative

selection. Yet, TE-related sites seem to be less conserved than non-TEs, and TE-derived binding sites also tend to be associated with less stable chromatin loops (less conservation between species and between cell types).

As TE activity has often been associated with the evolution of gene regulation, investigating of the link between TEs, chromatin loops, and gene regulation makes sense, and such a comparative genomics paper is timely (although comparative genomics on two species remains a challenge). Yet, I was not convinced by the manuscript, and I was even less convinced that the authors' findings should be communicated to a wide audience, due to the indirect and convoluted link between the data and the conclusions. My two main concerns are (i) the authors' interpretation of their results is largely speculative and is based on a vague "evolvability" argument that is never tested against more classical assumptions (neutral evolution, mutation bias...), and (ii) as a non-specialist, I found the reading extremely challenging. Details follow.

(i) Transposable elements are fascinating DNA sequences, not only because of their unique functional properties, but also by the fact that their presence and their distribution does not fit well into the traditional evolutionary synthesis. Understanding that genomes were dynamic was so important that the discovery of TEs granted the Nobel price to B. McClintock, with the underlying idea that TEs were "agents" that were reorganizing the genome or controlling gene expression in a somehow mysterious way. This idea collapsed in the 1980s with the first population genetics models of TE dynamics, showing that properties and distribution of transposable elements could be explained by the "Selfish DNA" theory -- replicative transposition is sufficient to ensure TE evolutionary success, and several independent lines of empirical evidence confirm that TEs are, in average, deleterious, and that various mechanisms have evolved in all organisms to actively silence transposition. In parallel, population genetics have shown that no known mechanisms could convincingly promote evolution towards more evolvability. Yet, this "selfish DNA" explanation was not considered as satisfactory by many, and countless effort has been spent in searching for a "purpose" of transposable elements. Of course, there might be cases where TE sequences and/or functions could be co-opted, but new or convoluted hypothesis should only be proposed after a formal dismissal of classical simple null hypotheses. For instance, the authors observe that ancient (prior to mouse-human divergence) TE-derived sites are underrepresented, and they conclude (if I understood properly) that this is because genomes are enriched in recent TE-derived sites that are actively contributing to adaptation in both species. It seems to me more parsimonious to assume that TE-derived sites might just be less conserved than non-derived sites (more neutral, or even more deleterious?). In another part of the manuscript, TE-derived sites are associated to loops that are more variable, both across species and across cell types within humans. A complex explanation involving TEs are diversity generators is proposed in the discussion, while again, the most straightforward explanation (at least, the one that should be discarded before proposing an adaptive explanation) is that these loops in which TEs are involved are more or less neutral and could be present or not without affecting the cell biology. In any case, for each observation, the authors should present a few straightforward evolutionary models which make different predictions, and test these predictions empirically, instead of proposing complex adhoc scenarios matching the data.

(ii) Comparing the reviews from specialists and non-specialists may probably help to determine whether the manuscript is suitable for a wide audience, but I have to report that trying to follow the authors' reasoning was particularly difficult to me. This can probably be partly explained by the lack of clear theoretical framework in which observations could be interpreted (most of the time, results are reported without explanation as to why a particular measurement was performed). I also have the feeling that the terminology is often vague, which does not help to get a precise picture of the results (e.g. line 178, "a large fraction", based on Fig 2E, corresponds to both ~ 20% (mouse) and ~ 75% (humans)). Some concepts (such as "latent CTCF binding sites") are not defined, and some conclusions are formulated in such a way that it is not easy to tell if they are actually informative (e.g. line 414, "TEs have, in fact, broadly affected CTCF binding throughout primate evolution": isn't it trivially the case with any genomic features? As TEs constitute ~ 45%

of the human genome, is there anything that is not broadly affected by TEs?). I also found some figures difficult to read and interpret, and I could not make sense of fig 6A and B (the caption states for 6A the vertical bars show the fraction of loops ordered by decreasing conservation, meaning that the left one is C and the right one is N0? What are the back dots?)

To exemplify the kind of effort it took me to translate the author's manuscripts in conclusions I would understand, let's focus on fig 4 and the corresponding text lines 263-279. The paragraph is entitled "TE activity has strongly contributed to chromatin looping in human and mouse" and presents a series of results on the evolutionary origin of TE and non-TE loop anchors. It actually took me quite some time to understand that the results simply tell that about 15% of anchors are TE-derived and loops arise randomly from any sequence. This predicts that $0.15 \times 0.15 = 2.3\%$ of loops should have 2 TE-derived anchors, 25.6% one, and 72.1% no TE anchors, which is almost exactly the reported proportions. Wouldn't "15% of loop anchors are TE-derived" simpler result than "TE activity has strongly contributed to..."? I could not do the same exercise for all figures and results, but my general feeling is that it is often possible to find a more straightforward and mainstream explanation to the results.

Detailed comments

- * I would add "chromatin" before "looping diversity" in the title.
- * Line 32, capital letter to Chromatin
- * Line 47, this sentence suggests that amplification could be a problem for class II transposons, which is absolutely not the case
- * Line 50, what are "innate" features of TEs?
- * Line 63: an unexpectedly large fraction of loops are anchored with TEs. Given that approx 15% of anchor loops are TE-derived and that $\sim 45\%$ of the genome is TE-derived, I am not sure about which direction would be "expected".
- * Line 74: TE proliferation, do you mean amplification or simply mobility?
- * Fig 1, I am not sure I found the example particularly clear. The authors have added the light blue lines to stress loop locations, but I am not convinced that anything special could be seen on the Hi-C map without the blue lines
- * Line 123 (but a bit everywhere in the ms): it is unclear if TE enrichment is relative to all TE-derived sequences or only from the consensus sequences. Conclusions would probably differ a lot depending on which sequence is considered.
- * Line 125, "TE-derived sites show a preference for...": this may look exactly what is expected if TE insertions were neutral and would drift independently across both species.
- * Line 173, caption of fig 2: what are "random repeats"? They don't seem to be a subsample of Enriched + Non-enriched repeats.
- * Line 175: I don't understand the p-value categories, two stars means exactly $p = 1.6 \times 10^{-14}$? This is an insanely significant p-value, why only two stars?
- * Line 218 "While extant copies of human-only ancestral TE types are detectable in the mouse genome" : it sounds totally contradictory to me, but I have the feeling that I did not understand what the authors meant by "human-only TEs", which does not seem to mean "human-specific".
- * Line 374: I am not sure I followed the reasoning. Is the histone marking in anchors TE specific?
- * Fig 5C, I share the authors surprise to the conservation of CTCF binding sites even in largely degraded sequences, and I suspect a bias in the selection of the sequence. I don't understand how the conservation score could be computed for the N0 class, as by definition there is non homologous region at the loop anchors?
- * Lines 417-422: again, this looks to me as a complicated way to say that there is no pattern, but I should have missed something.
- * Line 453: "It seems unlikely that these sites would exist within the TE consensus sequences absent a function to the TEs themselves": I don't understand this sentence.
- * Lines 461-470: I am not sure to follow the authors here, are several time scales seem to be merged (loops vary across cell types during development, among individuals in a population, and

change during species evolution, which dynamic scale is relevant to this discussion?).

* Line 480: "retrotransposon dispersals", do you mean "amplification"?

* Line 482, what is Mendelian selection, do you mean "Darwinian"?

* Line 483: the link between diversification and speciation seems even more speculative than the rest.

* Line 484, "TEs have broadly influenced CTCF binding", a similar idea is repeated several times in the ms but it is so vague that it could probably not be wrong.

* Line 486: explains much of the CTCF-binding variation: is it true? It concerns only 15% of the binding sites according to Fig 4.

* Line 628: even if the influence of computer programming in math notation becomes larger, the star is generally not considered as a proper notation for multiplication.

* Paragraph line 623: estimating the TE insertion age with this method only works if the TEs from this group / type follow a star-like phylogeny (one instant burst followed by an absence of transposition). If the phylogeny is more complicated and transposition is continuous through time, the consensus might not be at the root of the tree and age estimates are meaningless.

* Some supplementary figures were too large or encoded in a strange way, and I could not print them (12 Mb for 9 figures, starts to be heavy).

Signed: Arnaud Le Rouzic

Transposable elements strongly contribute to cell-specific and species-specific chromatin looping diversity in mammalian genomes.

RESPONSE TO REVIEWERS

Italics = original reviewer comments.

Red text = author responses.

We thank the reviewers for their detailed criticisms and suggestions for our manuscript. We have made substantial revisions to the manuscript to incorporate the suggestions made by both reviewers, including comparison of this work to the two recent studies noted by reviewer #1, and a broader treatment of possible interpretations of our results, with clearer explanations of why we believe our conclusion best fits the data we observe. We have also addressed the specific concerns of the reviewers below in a manner that we hope each of you find satisfactory.

Reviewer #1 (Remarks to the Author):

The evolution of genome organization is essential to understand the functions of higher-order chromatin structures. Mounting evidence indicates that mobile elements are a major contributor to transcription factor binding, thus this manuscript is tackling an important transcriptional regulation question. Overall, the authors have conducted careful bioinformatic analysis and integrated multiple dimension data. My major concern is the content overlap with two related studies. One is from Zhang et. al. recently published in Nature Genetics (Transcriptionally active HERV-H retrotransposons demarcate topologically associating domains in human pluripotent stem cells). The other is from Choudhary deposited in BioRxiv in Dec 2018 (Co-opted transposons help perpetuate conserved higher-order chromosomal structures). The authors need to clarify the novel findings in this manuscript that different from those two.

We are aware of both of these papers and agree it is important to clarify how our work builds on their results.

The first paper (Zhang et al.) mentioned by the reviewer presents a thorough analysis of how a single TE type has contributed to chromatin looping in human pluripotent cells. In particular, the authors show that HERV-H elements are enriched at the boundaries of TADs that are dynamic in the process of differentiation and experimentally demonstrate that elimination of these TE-derived boundaries alters TAD structure and disrupts regulatory control of genes necessary for maintenance of pluripotency. The key similarity between their results and ours is the association between TE-derived loop anchors and loops that are variable across cell types. We expand on their observations by showing that this phenomenon can be generalized across many TE types that, like HERV-H, contain CTCF binding motifs.

The second paper mentioned (in bioRxiv) presents a set of analyses very similar to our own. The authors investigate the contribution of TEs to looping in human and mouse cells, using some of the same cell types used in the present study, and demonstrate that TEs have made large contributions to loops that are conserved between human and mouse, with the key conclusion that TE-derived loop anchors have a stabilizing effect on higher-order chromatin structure. The key difference between their analysis and ours is that in they focused only on a highly-curated set of chromatin loops that are conserved between human and mouse lymphoblastoid cell lines. By doing so, they were able to interrogate the contribution of TEs to conserved looping with high confidence, but could not make any statements about the contribution of TEs to variable looping, which Zhang et al. show to be an important feature of TE-derived loop anchors. Rather than restricting to conserved or non-conserved loops, we chose to categorize loops into classes describing a spectrum of conservation levels across cells and species. Thus, we were able to interrogate the contribution of TEs to both conserved and variable loops, showing that their contributions to variable loops are proportionally larger than their contribution to conserved loops, consistent with the observations of Zhang et al.

In order to highlight similarities and differences between the current analysis and these two, we have made the following additions:

Line 102:

"These results expand on recent reports of TE enrichment at dynamic chromatin loop boundaries in human cells³⁸, showing that this phenomenon is generalizable across species and cell types. Furthermore, our findings complement reports that TE-derived CTCF sites contribute to conserved higher-order chromatin structures³⁹, demonstrating that over ~50% of species-specific loops in mouse and ~30% of cell-specific loops in human are TE-derived."

Line 292:

“the fractions we observed matched these expectations almost exactly: 25.1% of loops included 1 TE-derived anchor (Fig. 4A) and 2.6% were formed from two TE-derived anchors (Fig. 4B), and closely parallel previously reported TE-derived contributions to looping³⁹.”

Line 432:

“This suggests that the cell-specific properties of TE-derived loop anchors are persistent through evolutionary time, and correlates well with a recent study that identified TE enrichments at TAD boundaries that were dynamic over the course of cell differentiation³⁸.”

Line 516:

“This finding extends a recent report associating HERV-H elements with dynamic TAD boundaries in differentiating human cardiomyocytes³⁸, showing that the association with variability extends to loop anchors derived from TEs of many different types.”

Line 536:

“For instance, some TE-derived CTCF binding sites appear to make higher-order chromatin structures loops more robust to CTCF binding site turnover³⁹. We demonstrate that TEs make a proportionally larger contribution to variable loops, suggesting that the ability to readily adopt alternative chromatin conformations may also be selectively advantageous.”

Line 584:

“Our results complement those of Choudhary et al.³⁹, showing that TEs make important contributions to cell-specific and species-specific loops in addition to their possible functions in stabilizing conserved chromatin structure.”

Another concern is that even though K562 cells have been widely used to identify many interesting functional genomic findings. It is not a good cell system for the comparison conducted in this manuscript. Many translocations, fusions and segment duplication events have been detected in the genome of K562 cells. Those mutations can lead to different levels of artifacts in chromatin interactions and thus make the human-mouse comparison difficult to interpret. With many recent published Hi-C, ChIA-PET and Hi-ChIP data, the authors have better options.

We chose to use K562 in our analysis because we wanted to focus on closely-matched human and mouse cell lines. While ENCODE offers suitable data from many additional human cell lines, our choice of mouse datasets was very limited. We actually made an effort to include published mouse datasets from additional cell lines, but found the quality of available sequencing datasets to be insufficient for reliable comparisons. Therefore, we chose to restrict our analysis to human and mouse immune cells. That said, we recognize the importance of considering karyotypic abnormalities in comparing 3D structure across cell lines. We believe that K562 cells are a suitable cell line for this analysis based on four observations:

- 1) We observed close agreement in loop conservation results between pairwise comparisons of CH12 and GM12878 with those between CH12 and K562 (See Supplemental Table 7). While we saw a slight drop in conservation between CH12 and K562 compared to CH12 and GM12878, this likely reflects biological differences between the lymphoid and myeloid lineages.
- 2) Although K562 cells are near-triploid, this is not relevant to intrachromosomal looping and should not influence our results.
- 3) M-FISH and G-banding experiments show that K562 has a few large rearrangements and duplications, but is largely karyotypically normal (See Naumann, S., et al. (2001). *Leukemia Research*, 25(4), 313–322., Figs. 1 and 2).
- 4) The size of rearranged chromosomal segments in K562 far exceeds the average size of the loops used in this study and the number of loops observed in K562 exceeds the number of rearrangements by nearly an order of magnitude. We expect very few loops to span rearrangement breakpoints, thus rearrangements in K562 are unlikely to significantly skew our results.

One weakness of this manuscript is that unlike the two studies mentioned, there are no experimental validation in this study. Thus, the biological meanings is relative weak.

We acknowledge that this is a weakness of the current analysis. However, we are not currently set up to perform the types of experimental analyses needed for meaningful validation and we believe the time required to do so would have compromised the timeliness of these results. Accordingly, we opted to omit experimental validation from this analysis. However, experimental validations performed the two studies cited by the reviewer verify that transposable element insertions are sufficient to produce variable looping. Thus, we believe we can be confident in our conclusions even without experimental follow-up.

Minor points:

1) The author used CTCF binding sites generated by the standard ENCODE pipeline. Since non-uniquely mapped reads are automatically filtered in this pipeline. A large proportion ME-derived binding can be lost.

We now acknowledge this in the text and point out that our results represent a lower-bound estimate of the contribution of TEs to CTCF binding and chromatin looping as a consequence.

2) Can the authors provide the definition of "latent CTCF"? (line 183)

What we really meant was "CTCF motif" and should have been more precise in our terminology. We have rephrased the relevant sections and eliminated all uses of "latent CTCF."

3) The CTCF binding site gain and loss analysis were based on the whole peak regions. Due to the turn-over feature of TF, motif based analysis is more appropriate.

Without line references, we cannot be sure, but we believe this refers to the results presented in Fig. 3. We point out that the associated analyses were performed to answer the question of why certain TE types that are known to have been active in the MRCA of rodents and primates, are only enriched for CTCF binding only in human. The gain/loss analysis ruled out the possibility that the TE types in question were inserted into the human genome by amplifications after divergence (as the estimated divergence date and estimated TE ages are imprecise, this could not be shown only from the age distributions in Fig. 3A). Our preferred explanation was that the sequences were present in the MRCA, but they had lost the ability to bind CTCF in the mouse genome. Figure 3B confirms that the majority of these TE insertions were indeed present in the MRCA. As the reviewer rightly points out, analyzing the CTCF motifs present at these orthologous sites in both species is a more precise way of determining CTCF binding site turnover, which is why we performed the analysis presented in Fig. 3C: a comparison of CTCF binding motifs within the same loci. This confirms our hypothesis that CTCF binding motifs in mouse orthologs of these sequences have lost ability to strongly bind CTCF. We have attempted to clarify the relevant manuscript sections and figure legend to make this more apparent.

4) The hierarchy structure of TADs need to be considered when the author conduct the comparison.

We considered including a deeper analysis to investigate how variable TAD boundaries affect nested loops. For example, does moving a TAD boundary to an alternate anchor downstream affect the population of loops nested inside the TAD? Does changing a TAD in one region of the genome coincide with changes in TADs elsewhere? However, we felt this would add complexity to an already dense analysis. Nevertheless, this is an important point and we plan to revisit this in future analyses.

Reviewer #2 (Remarks to the Author):

Review of "Transposable elements strongly contribute to cell-specific and species-specific looping diversity in mammalian genomes", NSCOMM-19-20964.

In this paper, the authors study the evolutionary origins of the DNA sequences defining the anchors of some 3D chromatin loops. There is a growing interest in 3D chromatin structures, which role in genome dynamics and gene regulation seems to be important, and the questions raised in the manuscript can now be addressed thanks to the development of various technologies and databases reporting 3D structure information. Here, the authors focused on chromatin loops associated with the protein CTCF, which is also known to affect the transcription rate of neighboring genes. CTCF binds specific sequences in DNA, and these binding sites are often located in Transposable Elements.

The authors' results confirm that about 15% of loop anchor sites can be located in transposable elements in human and mouse. The paper also shows that canonical TE sequences frequently contains sequences corresponding to CTCF binding sites in both species. Some of these sites are evolutionary conserved since the mouse-human divergence, suggesting some kind of conservative selection. Yet, TE-related sites seem to be less conserved than non-TEs, and TE-derived binding sites also tend to be associated with less stable chromatin loops (less conservation between species and between cell types).

As TE activity has often been associated with the evolution of gene regulation, investigating of the link between TEs, chromatin loops, and gene regulation makes sense, and such a comparative genomics paper is timely (although comparative genomics on two species remains a challenge). Yet, I was not convinced by the manuscript, and I was even less convinced that the authors' findings should be communicated to a wide audience, due to the indirect and convoluted link between the data and the conclusions. My two main concerns are

(i) the authors' interpretation of their results is largely speculative and is based on a vague "evolvability" argument that is never tested against more classical assumptions (neutral evolution, mutation bias...),

We appreciate the feedback and have made numerous edits throughout the manuscript to clarify our core conclusions:

- 1) That TEs of all types (not just those that exhibit statistical enrichment for CTCF binding) have made major contributions to CTCF binding in human and mouse.
- 2) That TE-derived CTCF binding sites contribute to chromatin looping in proportion to their contribution to CTCF binding as a whole and does not depend on CTCF binding enrichment.
- 3) That TE-derived CTCF sites contribute more strongly to non-conserved and cell-specific loops than to loops that are conserved and/or stable across cell types.

The reviewer raises some excellent points about the modes of evolution that may have led to the data we observed. We have thus expanded our discussion to incorporate neutral evolution, biased mutation, and genetic drift inasmuch as they may have influenced the patterns we observed as much as was in keeping with the scope of this analysis.

and (ii) as a non-specialist, I found the reading extremely challenging. Details follow.

We have attempted to address each point raised by the reviewer, as well as making extensive edits throughout the manuscript to avoid the use of jargon and clarify our motivations and conclusions to make it as accessible as possible to non-specialist readers.

(i) Transposable elements are fascinating DNA sequences, not only because of their unique functional properties, but also by the fact that their presence and their distribution does not fit well into the traditional evolutionary synthesis. Understanding that genomes were dynamic was so important that the discovery of TEs granted the Nobel prize to B. McClintock, with the underlying idea that TEs were "agents" that were reorganizing the genome or controlling gene expression in a somehow mysterious way. This idea collapsed in the 1980s with the first population genetics models of TE dynamics, showing that properties and distribution of transposable elements could be explained by the "Selfish DNA" theory -- replicative transposition is sufficient to ensure TE evolutionary success, and several independent lines of empirical evidence confirm that TEs are, in average, deleterious, and that various mechanisms have evolved in all organisms to actively silence transposition. In parallel, population genetics have shown that no known mechanisms could convincingly promote evolution towards more evolvability.

We thank the reviewer for this excellent synopsis of the historical context of TE biology and evolution as it relates to host evolutionary pressures. It certainly gives a different perspective to our analysis and it was helpful in understanding and responding to the reviewer's comments and concerns.

Yet, this "selfish DNA" explanation was not considered as satisfactory by many, and countless effort has been spent in searching for a "purpose" of transposable elements. Of course, there might be cases where TE sequences and/or functions could be co-opted, but new or convoluted hypothesis should only be proposed after a formal dismissal of classical simple null hypotheses.

The reviewer's comments suggest to us that our results section, as originally written, included too much speculation on the underlying evolutionary mechanisms and consequences of our results, thus distracting from the results themselves. Accordingly, we now restrict these conclusions to the discussion section unless they are necessary to explain the motivation for follow-up analyses.

For instance, the authors observe that ancient (prior to mouse-human divergence) TE-derived sites are underrepresented, and they conclude (if I understood properly) that this is because genomes are enriched in recent TE-derived sites that are actively contributing to adaptation in both species. It seems to me more parsimonious to assume that TE-derived sites might just be less conserved than non-derived sites (more neutral, or even more deleterious?).

We take this as referring to the results presented in Fig. 3. As mentioned in our response to reviewer #1, we have substantially restructured this section of the manuscript to make our motivations and core conclusions much more clear, including removal of the statement which seems to have motivated this comment:

Original phrasing (Line 233 in original manuscript):

"It is possible that subsequent massive dispersals of mouse-specific B2 and B3 SINEs, which have no parallel in human evolution, have driven this loss of function by diverting CTCF binding from nearby ancestral sites, resulting in relaxed constraint."

As pointed out by the reviewer, our data do not directly support such a conclusion, and so this statement did not belong in the results section.

In regards to the reviewer's comment regarding the level of functional constraint on TE-derived sites, the absolute level of functional constraint on human or mouse orthologs is not important. Our result concretely shows a differential in the level of functional constraint between mouse and human loci, which supports our conclusion that these sites have lost CTCF binding function in mouse.

In another part of the manuscript, TE-derived sites are associated to loops that are more variable, both across species and across cell types within humans. A complex explanation involving TEs as diversity generators is proposed in the discussion, while again, the most straightforward explanation (at least, the one that should be discarded before proposing an adaptive explanation) is that these loops in which TEs are involved are more or less neutral and could be present or not without affecting the cell biology.

This is a valid observation, which we now acknowledge in the relevant sections. In fact, we do not believe that this possibility is inconsistent with our conclusions, in that Darwinian selection relies on the presence of neutral genetic variation at the population level. We have clarified in the relevant discussion section accordingly. That said, the underlying evolutionary constraint on TE-derived loop anchors in all conservation classes of loops suggests that these loops are indeed biologically relevant. However, investigating their exact biological functions, how these produce/relate to phenotypic variation, and whether they are selectively relevant, are beyond the scope of this analysis.

in any case, for each observation, the authors should present a few straightforward evolutionary models which make different predictions, and test these predictions empirically, instead of proposing complex adhoc scenarios matching the data.

We have attempted to include more discussion of alternative evolutionary hypotheses wherever possible, within the scope of the present analysis.

(ii) Comparing the reviews from specialists and non-specialists may probably help to determine whether the manuscript is suitable for a wide audience, but I have to report that trying to follow the authors' reasoning was particularly difficult to me. This can probably be partly explained by the lack of clear theoretical framework in which observations could be interpreted (most of the time, results are reported without explanation as to why a particular measurement was performed).

We have revised each results section to include a clear description of our motivations for running each analysis, and include relevant references to help frame and clarify analyses whenever possible.

I also have the feeling that the terminology is often vague, which does not help to get a precise picture of the results (e.g. line 178, "a large fraction", based on Fig 2E, corresponds to both ~ 20% (mouse) and ~ 75% (humans)). Some concepts (such as "latent CTCF binding sites") are not defined, and some conclusions are formulated in such a way that it is not easy to tell if they are actually informative (e.g. line 414, "TEs have, in fact, broadly affected CTCF binding throughout primate evolution").

These have been corrected in the revised manuscript and replaced by more precisely defined terms, using exact fractions or percentages wherever applicable.

isn't it trivially the case with any genomic features? As TEs constitute ~ 45% of the human genome, is there anything that is not broadly affected by TEs?.

The reviewer makes a valid point, but we do not believe this observation is incompatible with our first two core conclusions, as outlined above. Namely, the magnitude of TE-derived influence on CTCF binding and loop formation cannot be ignored, even if the fractional contribution does not represent a statistical enrichment of TEs among CTCF binding sites and/or loop anchors.

I also found some figures difficult to read and interpret, and I could not make sense of fig 6A and B (the caption states for 6A the vertical bars show the fraction of loops ordered by decreasing conservation, meaning that the left one is C and the right one is N0? What are the back dots?

We now reference the publication for the UpSet plot within the figure legend for those who have not encountered them before.

To exemplify the kind of effort it took me to translate the author's manuscripts in conclusions I would understand, let's focus on fig 4 and the corresponding text lines 263-279. The paragraph is entitled "TE activity has strongly contributed to chromatin looping in human and mouse" and presents a series of results on the evolutionary origin of TE and non-TE loop anchors. It actually took me quite some time to understand that the results simply tell that about 15% of anchors are TE-derived and loops arise randomly from any sequence. This predicts that $0.15 \times 0.15 = 2.3\%$ of loops should have 2 TE-derived anchors, 25.6% one, and 72.1% no TE anchors, which is almost exactly the reported proportions. Wouldn't "15% of loop anchors are TE-derived" simpler result than "TE activity has strongly contributed to..."?

The current revision includes a new presentation of the results that makes it clear that the observed fractions do, in fact, follow our expectations based on random interactions between TE-derived and native loop anchors.

I could not do the same exercise for all figures and results, but my general feeling is that it is often possible to find a more straightforward and mainstream explanation to the results.

Again, the difficulty seems to arise from our including too much speculation in the results section.

Detailed comments

* I would add "chromating" before "looping diversity" in the title.

Done

* Line 32, capital letter to Chromatin

Fixed.

* Line 47, this sentence suggests that amplification could be a problem for class II transposons, which is absolutely not the case

This sentence has been reworded to correct this.

* Line 50, what are "innate" features of TEs?

This section has been restructured to clarify which features of the TE sequences were investigated.

* Line 63: an unexpectedly large fraction of loops are anchored with TEs. Given that approx 15% of anchor loops are TE-derived and that ~ 45% of the genome is TE-derived, I am not sure about which direction would be "expected".

The reviewer makes a valid point in that we have not defined a null expectation nor rigorously tested for a significant deviation from that expectation. This was meant as a subjective evaluation of the results but we see how our original wording could be misleading and have changed the text accordingly. However, we do not believe the absence of a statistically significant deviation from a chance expectation for TE-derived loop frequency affects the core conclusions of the analysis.

* Line 74: TE proliferation, do you mean amplification or simply mobility?

Text has been revised to clarify that we mean primarily amplification.

* Fig 1, I am not sure I found the example particularly clear. The authors have added the light blue lines to stress loop locations, but I am not convinced that anything special could be seen on the Hi-C map without the blue lines

Based on the reviewer's wording, we are unsure whether the concern arises from a misunderstanding about the purpose of the blue lines or subjective interpretation of whether the two maps actually show biologically-meaningful differences.

To address the first possibility, we clarify that the blue lines do not highlight anything special on the map itself. Loops on each map are visible as clusters of dark red pixels against the lighter red background signal. We have emphasized the location of species-specific loops by enclosing these clusters in red or blue squares, with corresponding regions lacking a loop in the other species' map emphasized by enclosure within a circle of the same color. The X axis in our figure (which corresponds to the diagonal in a full Hi-C map) represents the noted segments of mouse chromosome 12 and human chromosome 14, and the blue lines simply clarify how points on the Hi-C map relate to relevant genomic features in the accompanying UCSC browser tracks.

To address the second possibility, we recognize that differences in Hi-C resolution and background can complicate map comparisons and make it difficult to discern species-specific features. We have tried to mitigate these

differences by using matched display resolution for both maps, and using matched normalization methods and parameters. We felt that it would be helpful to highlight relevant differential map features for readers not accustomed to reading Hi-C maps, but we can appreciate how the some may see these as distracting. We have included a version of figure 1 with this markup removed so the reviewer may draw his own conclusions as to whether a biologically meaningful difference exists.

* Line 123 (but a bit everywhere in the ms): it is unclear if TE enrichment is relative to all TE-derived sequences or only from the consensus sequences. Conclusions would probably differ a lot depending on which sequence is considered.

Enrichments in this section are calculated relative to all TE-derived sequences in the genome and we have clarified the text accordingly.

* Line 125, "TE-derived sites show a preference for...": this may look exactly what is expected if TE insertions were neutral and would drift independently across both species.

We have incorporated this topic in the revised Discussion section.

* Line 173, caption of fig 2: what are "random repeats"? They don't seem to be a subsample of Enriched + Non-enriched repeats.

These represent a random sampling of repeats from the DFAM database. We have clarified the text accordingly.

* Line 175: I don't understand the p-value categories, two stars means exactly $p = 1.6 \cdot 10^{-14}$? This is an insanely significant p-value, why only two stars?

As the reviewer points out, the important point is that all p-values are highly significant. To simplify interpretation, we now simply include a single asterisk for all p-values below the significance threshold of 0.05.

* Line 218 "While extant copies of human-only ancestral TE types are detectable in the mouse genome" : it sounds totally contradictory to me, but I have the feeling that I did not understand what the authors meant by "human-only TEs", which does not seem to mean "human-specific".

We have clarified definitions for these terms in the revised manuscript.

* Line 374: I am not sure I followed the reasoning. Is the histone marking in anchors TE specific?

The patterns of histone marks we observed are consistent with regulatory functions, and are not TE-specific. Rather, the point we are making is that both TE-derived and non-TE sites carry the same histone modification patterns, which is consistent with functional equivalency. This should be more clear in the revised manuscript.

* Fig 5C, I share the authors surprise to the conservation of CTCF binding sites even in largely degraded sequences, and I suspect a bias in the selection of the sequence. I don't understand how the conservation score could be computed for the N0 class, as by definition there is non homologous region at the loop anchors?

Functional constraint is measured using PhastCons 30-way (mouse) 46-way (human) placental mammal conservation scores. Even though N0 lacks sequence orthologs between human and mouse, many of these sequences share sequence orthologs with other species in the phylogeny used to calculate the PhastCons scores. Furthermore, the plots represent an average over all sites in each class such that the effects of sequences lacking an ortholog in any species will be minimal in the final plot. We now include additional references to phastCons at relevant points in the text and have clarified how they allow us to assess functional constraint for all classes of loops in the methods section.

* Lines 417-422: again, this looks to me as a complicated way to say that there is no pattern, but I should have missed something.

The proposed mechanism here is that strong CTCF motifs in TE consensus sequences may divert CTCF binding from nearby existing sites. The magnitude of the effects likely depends on the distance of the new site from the existing site, and may be selectively neutral. This point seems to have been confounded in the original discussion by also referencing how observed CTCF-binding enrichments may relate to this mechanism. We have made appropriate revisions.

* Line 453: "It seems unlikely that these sites would exist within the TE consensus sequences absent a function to the TEs themselves": I don't understand this sentence.

In the revised manuscript, we devote an entire paragraph in the discussion to the topic of how TEs might use CTCF as part of their survival strategy, which should serve to clarify our meaning.

* Lines 461-470: I am not sure to follow the authors here, are several time scales seem to be merged (loops vary across cell types during development, among individuals in a populations, and change during species evolution, which dynamic scale is relevant to this discussion?).

The revised manuscript clarifies how our results relate to early descriptions on the nature of TADs.

* Line 480: "retrotransposon dispersals", do you mean "amplification"?
Fixed.

* Line 482, what is Mendelian selection, do you mean "Darwinian"?
Fixed.

* Line 483: the link between diversification and speciation seems even more speculative than the rest.
As the reviewer points out, this is a highly speculative statement meant primarily to guide further discussion and hypothesis generation.

* Line 484, "TEs have broadly influenced CTCF binding", a similar idea is repeated several times in the ms but it is so vague that it could probably not be wrong.

Here we return to our earlier point that the bulk fraction of CTCF binding originating from TEs cannot be ignored even if there is no statistically significant enrichment of TEs among CTCF binding sites. We demonstrate that these sites have elevated levels evolutionary constraint and carry histone mark patterns consistent with regulatory function as further evidence that these CTCF binding sites have biologically-meaningful functions. Our evidence further demonstrates that anchoring chromatin loops appears to be prominent among those functions.

* Line 486: explains much of the CTCF-binding variation: is it true? It concerns only 15% of the binding sites according to Fig 4.

The relevant figure supporting this statement is Fig. 2A, which shows that >47% of mouse-specific binding sites and >30% of human-specific binding sites are TE-derived. In contrast, TE-derived CTCF sites comprise less than 10% of conserved CTCF binding sites. Accordingly, the new manuscript states: "Although many TE-derived CTCF binding sites are conserved in their placement between human and mouse, they contribute disproportionately to differentially-bound CTCF sites across cells and species (Fig. 2A) and the same trend is evident among variable loops across both species and cells (Fig. 6A-B)."

* Line 628: even if the influence of computer programming in math notation becomes larger, the star is generally not considered as a proper notation for multiplication.

We have reformatted the formula to avoid using stars to denote multiplication.

* Paragraph line 623: estimating the TE insertion age with this method only works if the TEs from this group / type follow a start-like phylogeny (one instant burst followed by an absence of transposition). If the phylogeny is more complicated and transposition is continuous through time, the consensus might not be at the root of the tree and age estimates are meaningless.

It is true that these effects may cause estimated ages for individual elements to deviate significantly from their actual ages (which are impossible to observe). However, because the consensus sequence represents the median configuration across the population, ages will be overestimated at the same frequency they are underestimated, thus having negligible impact on the age distribution as a whole. Because our conclusions are based on age distributions, not individual age estimates, this method is accurate enough for the purposes of this analysis.

* Some supplementary figures were too large or encoded in a strange way, and I could not print them (12 Mb for 9 figures, starts to be heavy).

We are sorry for the technical difficulties encountered by the reviewer. The necessity of including high-resolution figures had to be balanced against the maximum size requirement for the submission system. Although we were able to open and print the supplementary document on a variety of systems, we were, of course, unable to test it on every possible system configuration. The revised supplementary document includes rasterized versions of the largest figures. The result is a much smaller file size at the expense of some loss in resolution for the affected figures. We hope that this represents a reasonable compromise and resolves the issues.

Signed: Arnaud Le Rouzic

Reviewers' comments:

Reviewer #1 (Remarks to the Author):

The authors have addressed most of my questions and made the corresponding revision in the manuscript.

The biological function part remains to be a weak point for the whole story. How does genome-wide contribution of TEs to CTCF binding and chromatin organization, reported in the previous studies and here, affect transcriptional regulation? Without generating any new experimental data, the in silico validation needs to carry more weight. The current analysis on B0 class is vague and somehow confusing. The whole story will be much stronger if the authors can build more connections between species- / cell-specific loops or TADs to species-/cell-specific transcriptional regulation. Another potential way to improve is to generalize the recently published experiment-based findings with the results in this study. For example, are there any other HERV-H element derived CTCF TAD boundaries can potentially have similar effects on differentiation? This may be out of the scope of current manuscript, I will leave it to the authors to decide.

Reviewer #2 (Remarks to the Author):

Review of "Transposable elements strongly contribute to cell-specific and species-specific chromatin looping diversity in mammalian genomes", NSCOMM-19-20964A.

This is a revised version of a previous manuscript (NSCOMM-10-20964). Although the topic of the study was interesting and the analysis was thorough, I had two major concerns about the first version: (i) the interpretation of the results was mostly speculative, and (ii) the reading was challenging for a non-specialist.

In the revision, the authors have substantially modified the text (and marginally modified the figures) to address the reviewers' remarks. Reviewer #1's question about overlapping studies has been tackled seriously (I am myself a bit balanced about whether or not one should consider BioRxiv preprints as prior art); and reviewer #1's second point about the choice of K562 is now extensively discussed -- I am not competent to decide whether or not this weakens the conclusions of the paper.

In the same way, the authors have made a serious effort at addressing my detailed comments. However, the general points I raised from the previous version were about the way the manuscript was built, so it was probably unreasonable to expect an agreement after a round of revision. About my point (i), the authors have removed most of the evolutionary interpretations from the results section, and moved them in the discussion. This certainly makes the results section less dependent from a priori hypotheses on the underlying evolutionary mechanisms, although some vocabulary equivalences sound strange and not data-supported (for instance, "exapted" as an equivalent for "TE-derived"). Yet, the discussion remains very speculative, with an excess of "probability" adverbs (including "possibly", "may", "could", etc). For instance, line 535, "It is also possible that having a large pool of preferred and alternate CTCF binding sites from which to form chromatin loops may confer selective advantages", etc. I do not think such questions could be addressed by the kind of datasets examined here, which have little power to detect conservative selection, and even less for positive selection. Either the authors think their results actually support the hypothesis that some of these TE-derived sites have been under positive selection in one or both lineages (in which case, they should make a clear case out of it), either they think that a more straightforward interpretation (conservative selection for a minimum number of binding sites along the DNA) is not relevant, and this is the point that should be discussed before proposing alternative explanations.

About my second point (the difficulty in reading the paper), the authors have clarified or simplified some statements that I pointed out, but I still feel overwhelmed by the complexity of some results and the interpretation of some figures. Overall, the manuscript reads well if and only if the reader trusts the authors in their reasoning, and does not want to try to understand how the data actually support the results. There are many results in the manuscript that I was not able to review properly because I was not able to follow the authors' reasoning. For instance, lines 424-431, I found the claim that "this trend was driven partly by large, relatively recent expansions of enriched TE types in mouse, as evident by an increasing abundance of enriched TE types with decreasing conservation..." not evident at all. Fig 6B shows that less conserved sites have more mouse-enriched TEs in Mouse-Human comparisons (assuming that there is an error in figure caption in which C and D are swapped). Why would recent, mouse-specific enriched-TE insertions, over-represented in the N0 class? I don't mean that the authors' claim is wrong, but rather that the reasoning is too obscure and indirect to assess whether it is correct. Perhaps the authors meant that the observation was compatible with, or can be explained by the recent amplification? (which is radically different from bringing evidence). I had a similar feeling at so many places in the ms that it seems unreasonable to discuss all points individually. This is probably due to a combination of condensed results (many figures with small overcrowded panels), a genuinely complex analysis, a reviewer unfamiliar with jargon and graphical representations (UpSet figures?) in the field, and a complicated and oriented set of hypotheses to test.

* Minor details:

- abstract line 15: the coincidence between TE amplification and divergence remains dubious given the wide interval of divergence times from fig 6, and its evolutionary meaning remains unclear. I am not sure that highlighting this is particularly relevant.
- abstract line 16: "TE-derived looping variation may serve important roles in adaptive evolution", this again is largely speculative, and the claim is not supported by the data.
- line 131: "36 times more than expected by chance": do you mean that under a random model, TE-derived CTCF sites should represent only ~1.5% of mouse-specific CTCF sites? How is it calculated?
- line 143, "are robust and reproducible" is probably enough (76% does not sound "highly reproducible" to me...).
- line 209, Table 1 is duplicated with SupFig 1E. The four columns displaying P-values are probably meaningless. There is no explanation about the meaning of the numbers (number of copies and percentage of what?).
- line 284, in the rebuttal letter, the authors claimed that they have simplified the p-value starred summary, which is not the case here.
- line 285, what is a "native" site?
- line 494 and 551, there are duplicated sentences.
- line 561: parasitism is actually a form of symbiosis. Do you mean mutualism?
- lines 722 to 736, in this paragraph powers of ten are noted "e".
- SupFig 2, typo in the figure title ("associations")

Signed: Arnaud Le Rouzic

Black: Reviewer Comments

Red: Author Responses

Reviewers' comments:

Reviewer #1 (Remarks to the Author):

The authors have addressed most of my questions and made the corresponding revision in the manuscript.

The biological function part remains to be a weak point for the whole story. How does genome-wide contribution of TEs to CTCF binding and chromatin organization, reported in the previous studies and here, affect transcriptional regulation? Without generating any new experimental data, the in silico validation needs to carry more weight. The current analysis on B0 class is vague and somehow confusing. The whole story will be much stronger if the authors can build more connections between species- / cell-specific loops or TADs to species-/cell-specific transcriptional regulation.

We agree that this is an important area of validation and accordingly have added another section to the manuscript under the heading TE-derived variable loop anchors are associated with variable gene expression. In this section, we present a systematic analysis of the relationship between looping variability and variable target gene expression, demonstrating a significant correlation between variable gene expression and looping variation for both TE-derived and non-TE-derived chromatin loops. We also present a case study which extends the mechanisms recently demonstrated in:

Zhang, Y., et al., 2019. (n.d.). Transcriptionally active HERV-H retrotransposons demarcate topologically associating domains in human pluripotent stem cells. *Nature.com*.

showing that smaller-scale TE-induced looping variations (i.e., those that do not change overall TAD structure) may elicit gene expression changes by restricting the regulatory neighborhoods of their target genes, thus refining enhancer-promoter contacts.

Another potential way to improve is to generalize the recently published experiment-based findings with the results in this study. For example, are there any other HERV-H element derived CTCF TAD boundaries can potentially have similar effects on differentiation? This may be out of the scope of current manuscript, I will leave it to the authors to decide.

While this would be a very interesting direction for further investigation, we agree with the reviewer that it is outside the scope of the current manuscript. However, we believe the gene expression results we present provide sufficient evidence that the changes in gene looping variation we observe are biologically relevant and suggest that further investigation of individual looping changes can yield mechanistic insights into how these changes affect gene expression and, by extension, fitness and selection.

Reviewer #2 (Remarks to the Author):

Review of "Transposable elements strongly contribute to cell-specific and species-specific chromatin looping diversity in mammalian genomes", NSCOMM-19-20964A.

This is a revised version of a previous manuscript (NSCOMM-10-20964). Although the topic of the study was interesting and the analysis was thorough, I had two major concerns about the first version: (i) the interpretation of the results was mostly speculative, and (ii) the reading was challenging for a non-specialist.

We thank the reviewer for his thorough reading and attention to detail in reviewing the original and revised manuscript versions. Having the feedback of a non-specialist reader has been especially valuable to us in our efforts to make our results as accessible and intuitive as possible.

In the revision, the authors have substantially modified the text (and marginally modified the figures) to address the reviewers' remarks. Reviewer #1's question about overlapping studies has been tackled seriously (I am myself a bit

balanced about whether or not one should consider BioRxiv preprints as prior art); and reviewer #1's second point about the choice of K562 is now extensively discussed -- I am not competent to decide whether or not this weakens the conclusions of the paper.

In the same way, the authors have made a serious effort at addressing my detailed comments. However, the general points I raised from the previous version were about the way the manuscript was built, so it was probably unreasonable to expect an agreement after a round of revision.

We thank the reviewer for acknowledging our efforts to address his concerns while maintaining the overall structure and conclusions of the paper and will attempt to integrate his current suggestions as well.

About my point (i), the authors have removed most of the evolutionary interpretations from the results section, and moved them in the discussion. This certainly makes the results section less dependent from a priori hypotheses on the underlying evolutionary mechanisms, although some derived"). Yet, the discussion remains very speculative, with an excess of "probability" adverbs (including "possibly", "may", "could", etc).

This concern seems to reflect primarily a matter of stylistic preference. This area of research is still very new and we believe it is appropriate to make speculative statements in the discussion, provided they are consistent with the current state of knowledge and the present results. That said, the reviewer's point is well-taken and we have made efforts to clarify which statements are speculative (albeit still consistent with our results), and which are concrete conclusions with direct support from the data.

We address the specific concerns brought up by the reviewer below...

For instance, line 535, "It is also possible that having a large pool of preferred and alternate CTCF binding sites from which to form chromatin loops may confer selective advantages", etc. I do not think such questions could be addressed by the kind of datasets examined here, which have little power to detect conservative selection, and even less for positive selection. Either the authors think their results actually support the hypothesis that some of these TE-derived sites have been under positive selection in one or both lineages (in which case, they should make a clear case out of it), either they think that a more straightforward interpretation (conservative selection for a minimum number of binding sites along the DNA) is not relevant, and this is the point that should be discussed before proposing alternative explanations.

The reviewer makes a valid point that this is a highly speculative statement for which there is no direct support among our results. We refer back to our earlier assertion that it is appropriate to make such statements in the scope of the discussion section as long as they are qualified as such. We have extensively changed this discussion section to be absolutely clear that we are simply postulating as to what may explain the maintenance of such a large number of variable CTCF binding sites within the genome and now state explicitly that our methodology cannot discern between these different modes of selection.

About my second point (the difficulty in reading the paper), the authors have clarified or simplified some statements that I pointed out, but I still feel overwhelmed by the complexity of some results and the interpretation of some figures. Overall, the manuscript reads well if and only if the reader trusts the authors in their reasoning, and does not want to try to understand how the data actually support the results. There are many results in the manuscript that I was not able to review properly because I was not able to follow the authors' reasoning.

We appreciate the feedback and respond to each of the reviewer's concerns individually below. Without more specific feedback on exactly where we fail in explaining our reasoning, it is difficult for us to make broad changes to the manuscript in response. However, we have attempted to parse out the reviewer's broader concerns from the few examples below and have made changes accordingly throughout the manuscript to, we hope, improve the understandability of the analysis and the reasoning behind our conclusions.

For instance, lines 424-431, I found the claim that "this trend was driven partly by large, relatively recent expansions of enriched TE types in mouse, as evident by an increasing abundance of enriched TE types with decreasing conservation..." not evident at all. Fig 6B shows that less conserved sites have more mouse-enriched TEs in Mouse-Human comparisons (assuming that there is an error in figure caption in which C and D are swapped). Why would recent, mouse-specific enriched-TE insertions, over-represented in the N0 class? I don't mean that the authors' claim is wrong, but rather that the reasoning is too obscure and indirect to assess whether it is correct. Perhaps the authors

meant that the observation was compatible with, or can be explained by the recent amplification? (which is radically different from bringing evidence).

We have completely rewritten this section and made substantial changes to figure 6 to clarify our key conclusions:

- 1) Although a previous study has shown association between TEs and stability of conserved loops, we see a relatively larger contribution to looping variation.
- 2) The correlation with looping variability is seen in intraspecies comparisons between human cell types, not just across species, where neutral evolution, structural divergence (indels, rearrangements, etc.), and continued TE activity may easily explain such a trend.
- 3) While TE ages and associations with species-specific CTCF-enriched TE types and younger TEs do appear to at least partially explain this trend in cross-species comparisons, they do not explain the trend in intraspecies comparisons between human cells. Therefore, we can conclude that neutral drift and differential TE content/CTCF enrichment cannot entirely explain this trend.

I had a similar feeling at so many places in the ms that it seems unreasonable to discuss all points individually. This is probably due to a combination of condensed results (many figures with small overcrowded panels), a genuinely complex analysis, a reviewer unfamiliar with jargon and graphical representations (UpSet figures?) in the field, and a complicated and oriented set of hypotheses to test.

We have made our best efforts to pare out as much detail as possible from the figures and textual results to the bare minimum required to support our conclusions. We hope that the current revision allays some of these concerns. However, as the reviewer notes, this is a genuinely complex analysis.

* Minor details:

- abstract line 15: the coincidence between TE amplification and divergence remains dubious given the wide interval of divergence times from fig 6, and its evolutionary meaning remains unclear. I am not sure that highlighting this is particularly relevant.
- abstract line 16: "TE-derived looping variation may serve important roles in adaptive evolution", this again is largely speculative, and the claim is not supported by the data.

The two lines referenced above have been removed. We now close the abstract with a focus on our new gene expression results.

- line 131: "36 times more than expected by chance": do you mean that under a random model, TE-derived CTCF sites should represent only ~1.5% of mouse-specific CTCF sites? How is it calculated?

This is explained in the methods section, lines 729-735:

"The expected numbers of TE-derived human-specific and mouse-specific binding sites were calculated based on overlaps between species-specific CTCF binding sites and randomly selected windows following the size distribution of TE-derived CTCF binding sites in each species. We selected N random background regions from the given genome, where N is the number of species-specific CTCF sites derived from TEs. We then counted the number of overlaps between background regions and species-specific CTCF binding sites. We used the median number of overlaps observed over 1000 random trials as the expected number of TE/CTCF overlaps."

- line 143, "are robust and reproducible" is probably enough (76% does not sound "highly reproducible" to me...).

While beyond the scope of the current analysis, we have observed in our more-recent (unpublished) work that, even using the same enrichment test procedures, there is often variation in observed enrichments across different ChIP-seq datasets. This seems to trace largely to differences in ChIP-seq dataset quality, most notably the balance between completeness (i.e., saturation of true peaks in the genome) and false-positive peak content. Given our observations, we can confidently say that 76% is indeed highly reproducible.

- line 209, Table 1 is duplicated with SupFig 1E. The four columns displaying P-values are probably meaningless. There is no explication about the meaning of the numbers (number of copies and percentage of what?).

We have dropped table 1 as it is not critical to the results as presented in the main text. We retain the table in the supplemental figure. Additional clarification of the, counts, p-values, and percentages presented are now given in the supplemental figure legend.

- line 284, in the rebuttal letter, the authors claimed that they have simplified the p-value starred summary, which is not the case here.

We use two different statistical tests to assess significance here and needed to differentiate between which test was used for which comparison. In the current version, we replace ** with † for greater clarity.

- line 285, what is a "native" site?

"native site" and "native loop" are defined at the first uses:

- Line 62: "non-TE-derived loop anchors (native loop anchors)"
- Line 205: "non-TE binding sites (native sites)"

- line 494 and 551, there are duplicated sentences.

Fixed

- line 561: parasitism is actually a form of symbiosis. Do you mean mutualism?

We respect the reviewer's observation that the term "symbiosis" does not precisely imply our intended meaning. However, we note that symbiosis, in the popular vernacular, is commonly understood as a mutually-beneficial relationship while the precise definition of mutualism may not be universally understood by non-specialist readers. As such, we worry that rephrasing the sentence as:

"...they may actually exist as genomic mutualists rather than parasites."

would obscure our intended meaning and make the text less accessible to a broad audience.

- lines 722 to 736, in this paragraph powers of ten are noted "e".

Although we are used to seeing this notation in the scientific literature, we have changed all instances to $\times 10^{-n}$ notation.

- SupFig 2, typo in the figure title ("associations")

Fixed

Signed: Arnaud Le Rouzic

REVIEWERS' COMMENTS:

Reviewer #1 (Remarks to the Author):

No more questions.

Reviewer #2 (Remarks to the Author):

Review of "Transposable elements strongly contribute to cell-specific and species-specific chromatin looping diversity and variable gene regulation in mammalian genomes", NSCOMM-19-20964B.

This is the second revision of ms NSCOMM-19-20964. I have listed positive and less positive aspects of this work in my previous reviews, and will not detail them again. The manuscript is data-rich and proposes many analyses, figures, tables, and supplementary figures; going into every detail would hardly make sense. The authors have replied thoroughly to my previous comments, and we now have a short list of points of disagreement. Most of the reviewers' suggestions have been taken into account, in particular, I acknowledge the changes in the abstract and in fig 6. I think it is perfectly OK to have disagreements, and I do not see any reason to oppose publication.

In their response to the reviewers, the authors acknowledge that speculating on phenomena that are neither supported or rejected by the data is a matter of stylistic preference, which is basically true (authors remain free to decide the direction of the discussion). Yet, this is not neutral; statements like, line 684, "perhaps contributing to adaptability" cannot be rejected (after all, it may contribute to adaptability, whatever it means), but perhaps (and more probably) it may have little phenotypic consequences. I suspect that exposing exciting speculations is encouraged by the editorial line of high-profile journals, and I am fine with this as long as the authors, the editors, and the readers are aware that the published papers, in average, are enriched in erroneous claims compared to specialized journals.

Black: Reviewer Comments

Red: Author Responses

REVIEWERS' COMMENTS:

Reviewer #1 (Remarks to the Author):

No more questions.

We thank Reviewer #1 very much for their valuable feedback.

Reviewer #2 (Remarks to the Author):

Review of "Transposable elements strongly contribute to cell-specific and species-specific chromatin looping diversity and variable gene regulation in mammalian genomes", NSCOMM-19-20964B.

This is the second revision of ms NSCOMM-19-20964. I have listed positive and less positive aspects of this work in my previous reviews, and will not detail them again. The manuscript is data-rich and proposes many analyses, figures, tables, and supplementary figures; going into every detail would hardly make sense. The authors have replied thoroughly to my previous comments, and we now have a short list of points of disagreement. Most of the reviewers' suggestions have been taken into account, in particular, I acknowledge the changes in the abstract and in fig 6. I think it is perfectly OK to have disagreements, and I do not see any reason to oppose publication.

In their response to the reviewers, the authors acknowledge that speculating on phenomena that are neither supported or rejected by the data is a matter of stylistic preference, which is basically true (authors remain free to decide the direction of the discussion). Yet, this is not neutral; statements like, line 684, "perhaps contributing to adaptability" cannot be rejected (after all, it may contribute to adaptability, whatever it means), but perhaps (and more probably) it may have little phenotypic consequences. I suspect that exposing exciting speculations is encouraged by the editorial line of high-profile journals, and I am fine with this as long as the authors, the editors, and the readers are aware that the published papers, in average, are enriched in erroneous claims compared to specialized journals.

We agree with Reviewer #2 that it is perfectly acceptable to have disagreements. We certainly appreciate his perspective and offer many thanks for his time and attention to detail.